# Asymmetries in the Earth's dayside magnetosheath: results from global hybrid-Vlasov simulations

Lucile Turc[1], Vertti Tarvus[1], Andrew P. Dimmock[2], Markus Battarbee[1], Urs Ganse[1], Andreas Johlander[1], Maxime Grandin[1], Yann Pfau-Kempf[1], Maxime Dubart[1], and Minna Palmroth[1,3]

[1]Department of Physics, University of Helsinki, Helsinki, Finland
[2]Swedish Institute of Space Physics, Uppsala, Sweden
[3]Finnish Meteorological Institute, Helsinki, Finland

**Correspondence:** Lucile Turc (lucile.turc@helsinki.fi)

**Abstract.** Bounded by the bow shock and the magnetopause, the magnetosheath forms the interface between solar wind and magnetospheric plasmas and regulates solar wind-magnetosphere coupling. Previous works have revealed pronounced dawn-dusk asymmetries in the magnetosheath properties. The dependence of these asymmetries on the upstream parameters remains however largely unknown. One of the main sources of these asymmetries is the bow shock configuration, which is typically quasi-parallel on the dawn side and quasi-perpendicular on the dusk side of the terrestrial magnetosheath because of the Parker spiral orientation of the interplanetary magnetic field (IMF) at Earth. Most of these previous studies rely on collections of spacecraft measurements associated with a wide range of upstream conditions which are processed in order to obtain average values of the magnetosheath parameters. In this work, we use a different approach and quantify the magnetosheath asymmetries in global hybrid-Vlasov simulations performed with the Vlasiator model. We concentrate on three parameters: the magnetic field strength, the plasma density and the flow velocity. We find that the Vlasiator model reproduces accurately the polarity of the asymmetries, but that their level tends to be higher than in spacecraft measurements, probably because the magnetosheath parameters are obtained from a single set of upstream conditions in the simulation, making the asymmetries more prominent. A set of three runs with different upstream conditions allows us to investigate for the first time how the asymmetries change when the angle between the IMF and the Sun-Earth line is reduced and when the Alfvén Mach number decreases. We find that a more radial IMF results in a stronger magnetic field asymmetry and a larger variability of the magnetosheath density. In contrast, a lower Alfvén Mach number leads to a reduced magnetic field asymmetry and a decrease in the variability of the magnetosheath density, the latter likely due to weaker foreshock processes. Our results highlight the strong impact of the quasi-parallel shock and its associated foreshock on global magnetosheath properties, in particular on the magnetosheath density, which is extremely sensitive to transient quasi-parallel shock processes, even with the perfectly steady upstream conditions in our simulations. This could explain the large variability of the density asymmetry levels obtained from spacecraft measurements in previous studies.

# 1 Introduction

The interaction of the supermagnetosonic solar wind with the Earth's magnetosphere forms a standing bow shock which decelerates the incoming flow to submagnetosonic speeds in front of the obstacle. Extending between the bow shock and the magnetopause, the magnetosheath houses shocked solar wind plasma, which has been compressed and heated at the shock crossing. It is home to intense low-frequency wave activity, predominantly due to the mirror mode and the Alfvén ion cyclotron mode (e.g. Schwartz et al., 1996; Génot et al., 2009; Soucek et al., 2008). At the interface between the solar wind and the magnetosphere, the magnetosheath regulates the processes which transfer momentum and energy from the former to the latter and thus plays a key role in solar wind-magnetosphere coupling (Pulkkinen et al., 2016; Eastwood et al., 2017). Understanding and accurate modelling of this coupling therefore call for an in-depth knowledge of magnetosheath properties and their dependence on upstream solar wind parameters.

Since the early gasdynamic model of Spreiter et al. (1966), the magnetosheath has been subject to intensive scrutiny (e.g. Petrinec et al., 1997; Paularena et al., 2001; Longmore et al., 2005; Lucek et al., 2005; Dimmock and Nykyri, 2013; Lavraud et al., 2013; Dimmock et al., 2017). These studies revealed that the magnetosheath properties display significant spatial variations, as a function of the distance from the boundaries, with for example the formation of the plasma depletion layer near the magnetopause during northward interplanetary magnetic field (IMF) conditions (e.g. Zwan and Wolf, 1976; Wang et al., 2004), and as a function of the distance from the Sun-Earth line, with pronounced dawn-dusk asymmetries (see the reviews by Walsh et al., 2014; Dimmock et al., 2017, and references therein). One of the main sources of these dawn-dusk asymmetries is the bow shock, another one being the leakage of magnetospheric particles into the magnetosheath, which result in a dawn-dusk asymmetry of the energetic ion and electron components in the magnetosheath plasma (Anagnostopoulos et al., 2005; Cohen et al., 2017). In this paper, we concentrate on the impact of the bow shock properties on the large-scale distribution of the magnetosheath properties.

The shock properties depend strongly on the angle $\theta_{\mathrm{Bn}}$ between the IMF and the local normal to the shock's surface. Because of the Parker-spiral orientation of the IMF at Earth, which makes a $45°$ angle with the Sun-Earth line, the dusk side of the magnetosheath generally lies downstream of a quasi-perpendicular ($Q_\perp$) shock ($\theta_{\mathrm{Bn}} > 45°$), while the dawn side is associated with a quasi-parallel ($Q_\parallel$) shock ($\theta_{\mathrm{Bn}} < 45°$). Even in the fluid approximation, these contrasted shock regimes result in different plasma properties in the downstream region. Using the Rankine-Hugoniot jump conditions, Walters (1964) found larger plasma densities and temperatures downstream of the quasi-parallel shock than downstream of the quasi-perpendicular shock. Global magnetohydrodynamic (MHD) simulations have brought additional support to the dawn magnetosheath being home to a hotter and denser plasma, while the magnetic field strength and flow velocity are larger on the dusk flank (Samsonov et al., 2001; Walsh et al., 2012).

Investigating magnetosheath asymmetries using spacecraft measurements is a challenging task because it requires an extensive spatial coverage of this region. Since simultaneous measurements in different parts of the magnetosheath are scarce, most observational studies rely on compilations of spacecraft observations from different passes through this region to build statistical maps of the magnetosheath properties (Paularena et al., 2001; Němeček et al., 2002; Longmore et al., 2005; Walsh

et al., 2012; Dimmock and Nykyri, 2013). The main drawbacks of this approach are that these data are collected during vastly different upstream conditions and that the position of the spacecraft relative to the magnetosheath boundaries is essentially unknown. The former issue is generally addressed by normalising the magnetosheath parameters with their solar wind counterparts, while empirical models of the magnetosheath boundaries provide an estimate of the relative position of the spacecraft inside the magnetosheath.

Consistent with the aforementioned theoretical and numerical works, observational studies have reported a dusk-favoured asymmetry of the magnetic field strength and of the plasma velocity (Longmore et al., 2005; Walsh et al., 2012; Dimmock and Nykyri, 2013; Dimmock et al., 2017). The ion temperature, on the other hand, showcases a dawn-favoured asymmetry, probably due to enhanced heating at the more turbulent quasi-parallel shock (Walsh et al., 2012; Dimmock et al., 2015a). Furthermore, magnetic field and velocity fluctuations are stronger in the dawn magnetosheath (Dimmock et al., 2014, 2016a), while temperature anisotropy and mirror mode wave activity are more prominent in the dusk sector (Dimmock et al., 2015b; Soucek et al., 2015). In an earlier study by Tátrallyay and Erdős (2005), no dawn-dusk asymmetry was evidenced for mirror mode occurrence, but it should be noted that the data were not organised according to the shock configuration in this work, contrary to the more recent studies by Dimmock et al. (2015b) and Soucek et al. (2015).

The density asymmetry turned out to be more elusive in spacecraft measurements. Though a clear dawn-favoured asymmetry was found in several data sets (Paularena et al. (2001) for solar maximum; Němeček et al. (2002); Walsh et al. (2012); Dimmock et al. (2016b)), others did not display any significant asymmetry levels (Dimmock and Nykyri, 2013; Paularena et al., 2001, for solar minimum), or even an asymmetry with a changing polarity depending on the location inside the magnetosheath (Němeček et al., 2003; Longmore et al., 2005). We note that because they originate from different spacecraft missions, the data sets used in these studies cover various parts of the magnetosheath: nightside (Paularena et al., 2001), close to the terminator (Němeček et al., 2002, 2003), dayside at high latitudes (Longmore et al., 2005), and dayside near the equatorial plane, either near the magnetopause (Walsh et al., 2012; Dimmock et al., 2016b) or across the whole magnetosheath thickness (Dimmock and Nykyri, 2013). They also correspond to various parts of the solar cycle, which may affect the level of the density asymmetry because the average solar wind parameters depend on solar activity. It is noteworthy than Paularena et al. (2001) and Dimmock et al. (2016b) reported opposite behaviours of the density asymmetry as a function of the solar cycle.

The dependence of magnetosheath asymmetries on upstream parameters can bring insight into the processes that create them. Longmore et al. (2005) and Dimmock et al. (2017) found no clear dependence of the density and velocity asymmetries on the IMF direction, suggesting that they may not be driven by the bow shock. On the other hand, the level of these asymmetries increases with the Alfvén Mach number ($M_A$), as does the temperature asymmetry, according to the numerical simulations performed by Walsh et al. (2012). They also show that an increasing $M_A$ would also tend to increase the magnetic field strength asymmetry. Walsh et al. (2012) ascribe the observed density asymmetry to the asymmetric bow shock shape, as its quasi-parallel sector lies closer to the magnetopause than its quasi-perpendicular sector. They argue that the apparent lack of dependence of the density asymmetry on the IMF direction in statistical studies is likely due to the limited number of data points associated with non-Parker-spiral IMF orientations. As evidenced by these contradicting claims, many open questions

remain regarding the precise sources of the observed magnetosheath asymmetries and their dependence on upstream solar wind conditions.

Asymmetries in the magnetosheath parameters result in turn in an asymmetric magnetospheric driving. Large amplitude velocity fluctuations in the magnetosheath are conducive to a faster growth of the Kelvin-Helmholtz instability at the Earth's magnetopause and larger plasma transport through the boundary (Nykyri et al., 2017). Such velocity fluctuations are stronger in the quasi-parallel magnetosheath (Dimmock et al., 2016a), and these, accompanied with the lower tangential field strength in this region, result in the Kelvin-Helmholtz instability favouring the quasi-parallel flank (Henry et al., 2017). Also, ions of magnetosheath origin in the plasma sheet present a dawn-favoured asymmetry of about $30 - 40\%$ (Wing et al., 2005). This asymmetry could partially be explained by the temperature asymmetry in the magnetosheath, while additional heating processes may be regulated by the asymmetric distribution of other magnetosheath parameters (Dimmock et al., 2015a; Dimmock et al., 2017).

Numerical simulations can help shed new light onto magnetosheath asymmetries, as they provide a global view of the magnetosheath for a given set of solar wind conditions, instead of relying on statistical maps constructed from measurements associated with a variety of upstream parameters. This also removes possible errors when determining the context of magnetosheath measurements, which must be combined with time-lagged data from an upstream monitor in observational studies. To date, most numerical studies of magnetosheath asymmetries have used MHD models (Walsh et al., 2012; Dimmock and Nykyri, 2013), though the temperature asymmetry was qualitatively compared with the outputs from a hybrid-Particle-in-Cell simulation by Dimmock et al. (2015a). The physics of the quasi-parallel bow shock and its associated foreshock are however inherently kinetic in nature, and thus a kinetic approach is better suited to study magnetosheath parameters downstream of the quasi-parallel shock.

Hybrid-kinetic simulations, that is, including ion kinetic effects but where electrons are treated as a fluid, are extensively used to study the interaction of the solar wind with planetary magnetospheres, and in particular foreshock, bow shock and magnetosheath processes (e.g. Omidi et al., 2005; Lin and Wang, 2005; Blanco-Cano et al., 2006; Omidi et al., 2014; Karimabadi et al., 2014; Turc et al., 2015; Modolo et al., 2018; Palmroth et al., 2018). A number of numerical studies of the magnetosheath focus on wave activity in this region and the competition between mirror modes and Alfvén ion cyclotron waves (e.g., Trávníček et al., 2007; Herčík et al., 2013; Hoilijoki et al., 2016). The numerical simulations of Omidi et al. (2014) revealed large-scale filamentary structures in the quasi-parallel magnetosheath, while Karimabadi et al. (2014) investigated small-scale processes such as turbulence and reconnection.

In this paper, we present the first analysis of magnetosheath asymmetries as obtained from global ion kinetic simulations performed with the hybrid-Vlasov model Vlasiator (von Alfthan et al., 2014; Palmroth et al., 2018). We use a set of three different runs to investigate the effects of the IMF cone angle $\theta_{\mathrm{B}x}$ (measured between the IMF vector and the Sun-Earth line) and the solar wind Alfvén Mach number, which are key parameters controlling the shock properties. In this first study based on hybrid-Vlasov simulations, we choose to focus on three primary magnetosheath parameters: the magnetic field strength $B$, the plasma velocity $V$ and the ion density $n_{\mathrm{p}}$. For the latter, we will attempt to identify possible reasons for its large variability in observational studies.

**Table 1.** Summary of the run parameters

| Run name | Simulation plane | $\Delta r$ [km] | IMF cone angle $\theta_{\mathrm{B}x}$ | IMF strength [nT] | $M_A$ | $n_{\mathrm{SW}}$ $[\mathrm{cm}^{-3}]$ | $\mathbf{V}_{\mathrm{SW}}$ $[\mathrm{km\,s}^{-1}]$ |
|---|---|---|---|---|---|---|---|
| Run 1 | $x-z$ plane | 300 | $45°$ | 5 | 6.9 | 1 | $(-750,0,0)$ |
| Run 2A | $x-y$ plane | 227 | $30°$ | 5 | 6.9 | 1 | $(-750,0,0)$ |
| Run 2B | $x-y$ plane | 227 | $30°$ | 10 | 3.5 | 1 | $(-750,0,0)$ |

## 2 Methodology

### 2.1 The Vlasiator simulation

Vlasiator is a hybrid-Vlasov model designed to perform global simulations of the Earth's plasma environment while retaining ion kinetic physics (von Alfthan et al., 2014; Palmroth et al., 2018). In the hybrid-Vlasov formalism, ions are treated as velocity distribution functions evolving in phase space whereas electrons are modelled as a cold massless charge-neutralising fluid. The temporal evolution of the system is obtained by solving Vlasov's equation, coupled with Maxwell's equations. Ohm's law, including the Hall term, provides closure to the system. In Vlasiator, the use of realistic proton mass and charge, together with the full strength of the Earth's dipole field, results in processes being simulated at their actual physical scales, as encountered in near-Earth space. This makes the comparison with spacecraft measurements straightforward.

The runs presented in this paper are two-dimensional (2D) in ordinary space. Each grid cell in ordinary space is self-consistently coupled with a 3D velocity space in which the ion distribution functions evolve. In each ordinary space cell, the plasma parameters are obtained as the moments of the velocity distribution function, by integration over the velocity space. The coordinate system used in the simulation is equivalent to the Geocentric Solar Ecliptic (GSE) reference frame. In this Earth-centred frame, the $x$-axis points towards the Sun, $z$ is perpendicular to the Earth's orbital plane and points northward, and $y$ completes the right-handed triplet. Depending on the runs, the simulation domain covers either the equatorial ($x-y$) or the noon-midnight meridional ($x-z$) plane (see Table 1 for a summary of the run parameters). In equatorial runs, we use the Earth's magnetic dipole with its actual value of $8.0 \times 10^{22}\,\mathrm{A\,m}^2$, while for runs in the noon-midnight meridional plane, a 2D line dipole is used (Daldorff et al., 2014). In all runs, the solar wind flows into the simulation domain from the $+x$ edge. Copy conditions are applied at the other walls of the simulation domain, while periodic conditions are employed for the out-of-plane cell boundaries (i.e., in the $z$ direction for a run in the $x-y$ plane). The inner boundary of the simulation domain is a circle at about $4.7\,\mathrm{R_E}$ from the Earth's centre, considered a perfect conductor.

### 2.2 Runs used

In this study, we analyse three Vlasiator runs, each corresponding to different IMF conditions (see Table 1). This allows us to investigate the influence of the IMF orientation and strength (and by extension the Alfvén Mach number) on magnetosheath asymmetries. In all three runs, the solar wind ions are injected at the inflow boundary as a Maxwellian population with a density

$n_{\mathrm{SW}} = 1\,\mathrm{cm}^{-3}$ and a temperature $T_{\mathrm{SW}} = 0.5\mathrm{MK}$, flowing at a velocity $\mathbf{V}_{\mathrm{SW}} = (-750, 0, 0)\,\mathrm{km\,s}^{-1}$, thus corresponding to fast solar wind conditions.

In the reference run, hereafter Run 1, the IMF vector makes a $45°$ cone angle with the Sun-Earth line and lies in the $x-z$ plane, with $\mathbf{B} = (3.54, 0., -3.54)\,\mathrm{nT}$. This results in an Alfvén Mach number $M_A = 6.9$ and a magnetosonic Mach number $M_{\mathrm{ms}} = 5.5$, which fall inside the range of typical values for these Mach numbers at Earth (Winterhalter and Kivelson, 1988). Therefore, despite the large solar wind speed in our runs, we have a typical density compression ratio at the bow shock with our input parameters. The simulation domain extends from $-48.6$ to $64.3$ Earth radii ($R_E = 6371$ km) in the x direction and from $-59.6$ to $39.2\,R_E$ in the z direction. The spatial resolution in this run is $\Delta r = 300\,\mathrm{km}$, that is, 1.3 solar wind ion inertial length ($d_i = 227.7$ km), and the velocity space resolution is $30\,\mathrm{km/s}$. In a hybrid-Vlasov simulation, these resolutions are sufficient to resolve the dominant ion kinetic processes in the foreshock-bow shock-magnetosheath system (see Hoilijoki et al., 2016; Pfau-Kempf et al., 2018; Dubart et al., 2020). Possible limitations due to the chosen resolutions are discussed in Section 4.

Run 1 simulates the noon-midnight meridional plane of near-Earth space, as it was initially designed to study e.g. dayside and nightside reconnection in the presence of the foreshock (Hoilijoki et al., 2019). For an Alfvénic Mach number $M_A = 6.9$ as in Run 1, the quasi-perpendicular portion of the bow shock lies roughly at the same distance from Earth both in the $x-y$ and the $x-z$ planes, while its quasi-parallel sector is found closer to Earth, according to MHD simulations (Chapman et al., 2004). Therefore, if the IMF lies in the $x-z$ plane, the position and shape of the bow shock in this plane are essentially the same as those observed in the equatorial plane for an IMF vector in the $x-y$ plane. Since the main parameter controlling most magnetosheath asymmetries is the bow shock configuration (Dimmock et al., 2017), the IMF configuration in Run 1 is roughly equivalent to a Parker spiral IMF orientation in the equatorial plane in terms of bow shock and outer magnetosheath properties (i.e. away from the cusps and the reconnecting magnetopause). Although this setup is not ideal, it is sufficient for the purpose of the present study, and we deemed that running a new simulation was not warranted, as Vlasiator runs are computationally expensive, requiring of the order of several million CPU-hours. We will use this run as a reference for the most typical IMF orientation at Earth.

The other set of two runs, Runs 2A and 2B, are equatorial runs, with a $30°$ cone angle IMF in the $x-y$ plane. In both Runs 2A and 2B, the simulation box extends from $-7.9$ to $46.8\,R_E$ in the x direction and between $\pm 31.3\,R_E$ in the y direction. The spatial resolution is $\Delta r = 227\,\mathrm{km}$, that is, 1 solar wind ion inertial length, and the velocity space resolution is $30\,\mathrm{km/s}$. In Run 2A, the IMF strength is set to 5 nT, as in Run 1, while in Run 2B, its value is set to 10 nT. As a result, the Alfvén Mach number $M_A$ is reduced to 3.5 in this run, half of its value in Runs 1 and 2A where $M_A = 6.9$. To avoid confusion in the case where the simulation plane is not the equatorial plane, we will refer to the polarity of the magnetosheath asymmetries as $Q_\perp$-favoured or $Q_\parallel$-favoured, instead of the dawn-dusk terminology generally used in observational studies.

## 2.3 Analysis method

In each run, we divide the dayside magnetosheath into sectors within which we calculate the average magnetosheath properties, as illustrated by the black curves in Figure 1a. Determining the exact bow shock and magnetopause positions proved to be rather impractical, as their position can vary significantly depending on the parameter which is selected to define the boundary

(Palmroth et al., 2018; Battarbee et al., 2020). Therefore, we decided to use a simpler method to define approximate boundaries that would serve as the inner and outer limits for our magnetosheath binning. We use for simplicity the same shape as that of the Shue et al. (1997) magnetopause model (of the form $r = r_0(2/(1+\cos\theta))^\alpha$), where $r_0$ is the stand-off distance, $\theta$ the angle from the Sun-Earth line and $\alpha$ the flaring parameter, to delineate the boundaries of the bins in the radial direction. This shape approximates relatively well the bow shock and magnetopause shape in our simulations when different flaring parameters are used. For the bow shock, we also use a different flaring parameter for the quasi-parallel and the quasi-perpendicular flanks, to account for the asymmetric bow shock shape.

For each run, the values for $r_0 = r_{\min}$ (inner boundary), $r_0 = r_{\max}$ (outer boundary) and $\alpha$ are selected by visual inspection so as to maximise the coverage of the magnetosheath while remaining sufficiently far from the bow shock and the magnetopause to avoid including data from other regions. The two intermediate radial boundaries are placed at one third and two thirds of the magnetosheath thickness $r_{\max} - r_{\min}$. We denote the relative position between the magnetosheath boundaries as $F_{\mathrm{Msheath}} = (r - r_{\min})/(r_{\max} - r_{\min})$. In the azimuthal direction, the magnetosheath is divided into 18 $10°$-wide angular bins. In our analysis, we will only focus on the central and outer sets of radial bins, to ensure that the cusps are excluded and that magnetopause processes do not affect our results in Run 1.

Inside each of these bins, we calculate the average values of various magnetosheath parameters, namely the ion density, the plasma bulk velocity and the magnetic field strength. In addition to spatial averages within each bin, we also perform temporal averages in order to minimise the effects of transient features originating from the foreshock or arising inside the magnetosheath. Here we use $150$ s temporal averages to calculate the magnetosheath parameters, which was found as a good trade-off to remove the effect of transients with only limited changes in the position of the magnetosheath boundaries. This averaging interval is much larger than the proton gyroperiod in the solar wind ($13$ s in Runs 1 and 2A and $6.5$ s in Run 2B), and is comparable with the $180$ s window used by Dimmock et al. (2017) for spacecraft measurements. We also calculated the median value of the magnetosheath parameters within each bin, for the same spatial and temporal sample, and we obtained very similar results. To facilitate the comparison with the most recent studies of magnetosheath asymmetries (Dimmock et al., 2017, and references therein), which are based on average values, we present here the results obtained from averaging the magnetosheath parameters. As in Dimmock et al. (2017), we estimate the error associated with each parameter within each bin as the standard error of the mean, $SEM = \sigma/\sqrt{N}$, where $\sigma$ is the standard deviation and $N$ is the number of simulation cells inside each bin.

We note here that because of the 2D set-up of our simulations, field lines tend to pile-up at the magnetopause, as they cannot slip along the magnetosphere flanks. As a result, the bow shock moves slowly outwards. To ensure that the comparison of the different runs is meaningful, we select time intervals in Run 1 and Run 2A when the bow shock shape was comparable, as it should not be strongly affected by the different IMF cone angles. In Run 1, we calculate the average magnetosheath parameters between $t = 700$ and $t = 850$ s, when the simulation has properly initialised and before the onset of intense dayside reconnection, which could cause changes in the flow pattern near the magnetopause, and to limit the effects of reconnection-driven magnetic islands in the magnetosheath (Pfau-Kempf et al., 2016). In Runs 2A and 2B, we use the interval from $t = 350$

to $t = 500$ s. The initialisation phase of these runs is shorter than in Run 1 because of their smaller simulation domain. At these times, all three runs have reached a quasi-steady state.

Following Dimmock et al. (2017), we define the asymmetry of the magnetosheath parameters as:

$$A = 100 \times \left( \frac{Q_\perp - Q_\parallel}{Q_\perp + Q_\parallel} \right) \tag{1}$$

where $Q_\perp$ is the average value of a magnetosheath parameter (here magnetic field strength, plasma velocity or ion density) in a given azimuthal and radial bin in the quasi-perpendicular magnetosheath, and $Q_\parallel$ its average value in the corresponding opposite bin, i.e., symmetric with respect to the Sun-Earth line, in the quasi-parallel magnetosheath. The error of the asymmetry is estimated as the extreme values of $A$ when injecting $Q_\perp \pm SEM$ and $Q_\parallel \pm SEM$ into Eq. 1 (see Dimmock et al., 2017). Note that we use the same arrangement of quasi-perpendicular and quasi-parallel bins in the analysis of Runs 2A and 2B, even though the reduced cone angle in these runs shifts the transition between the two shock regimes away from the bow shock nose. This facilitates the comparison with observational studies, which do not account for the IMF cone angle in their mapping of the magnetosheath parameters (e.g. Dimmock et al., 2017).

We also note here that although the simulation input parameters deviate from average values in the solar wind at Earth, this is not an issue for the comparison with previous observational studies. The statistical data sets, based on compilations of magnetosheath measurements associated with a wide variety of solar wind conditions, rely on the assumption that magnetosheath parameters can be normalised to their solar wind counterparts to obtain the average distribution of magnetosheath properties. In the present work, the normalisation of the data to the solar wind quantities together with the typical shock Mach numbers and compression ratio in our simulations ensure that the comparison with spacecraft observations is meaningful.

## 3 Results

### 3.1 Magnetic field strength

Colour-coded in the top panels of Figure 1 is the magnetic field strength in the dayside magnetosheath and the neighbouring regions, normalised to the IMF strength, in each of the three runs. As indicated by the magnetic field lines (light grey curves), the quasi-parallel sector of the bow shock and its associated foreshock extend in the lower part of each plot, upstream of the southern ($z < 0$, in Run 1) or dawnside ($y < 0$, in Runs 2A and 2B) magnetosheath. The colour scheme is chosen to highlight the areas of the magnetosheath where the normalised magnetic field strength is above or below 4, which is the upper limit for the magnetic field compression at the bow shock crossing according to the Rankine-Hugoniot jump conditions (Treumann, 2009). In Run 2B, the normalised magnetic field strength is below 4 in most of the magnetosheath, due to the weaker compression at the bow shock when the Alfvén Mach number is low. In Runs 1 and 2A, it remains below 4 in the first few $R_E$ downstream of the subsolar bow shock, and in a much broader area in the flank magnetosheath. In regions closer to the magnetopause, its values increase well above 4 due to the field lines piling up in front of the magnetosphere. In the subsolar region, the effects of pile-up are visible even in the outermost magnetosheath bins used in our study (black curves), while they are limited to the central and inner magnetosheath bins further on the flanks. They also extend further out in the quasi-perpendicular

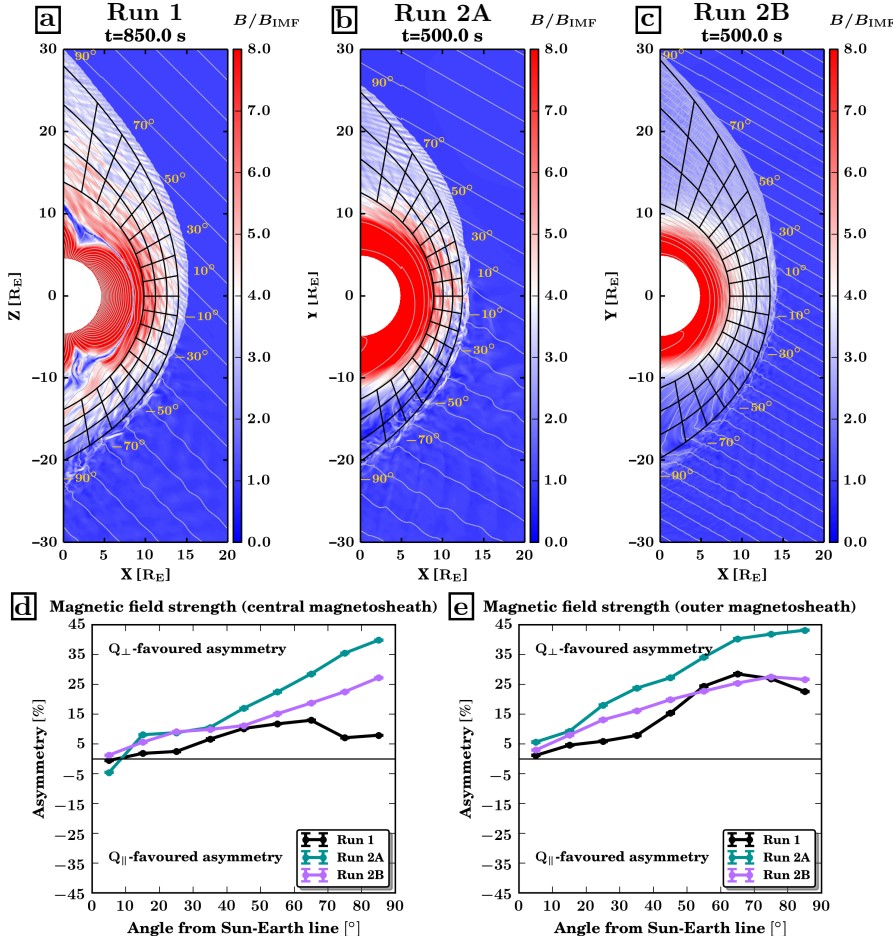

**Figure 1.** Top panels: magnetic field strength in the simulation plane, normalised with the IMF strength, in Run 1 at time $t = 850\,\text{s}$ (a), in Run 2A (b) and 2B (c) at time $t = 500$ s. The light grey lines show magnetic field lines. The spatial bins used to calculate the average magnetosheath parameters are shown in black. Bottom panels: magnetic field strength asymmetry in the central (d) and outer (e) magnetosheath. The error bars are obtained from the extreme values of the asymmetry based on the standard error of the mean in each bin (see Section 2.3).

magnetosheath than downstream of the quasi-parallel shock, due to the IMF orientation. Similar features due to pile-up are also observed in the statistical maps compiled by Dimmock et al. (2017) (see the top panels of their Figure 5.1). The only significant difference between our simulation results and Dimmock et al. (2017)'s maps is the large magnetic field strength along the northern magnetopause close to the terminator in Run 1, which is likely due to the 2D set-up of our simulation, resulting in enhanced field line pile-up. In the following, we will exclude from our analysis the innermost magnetosheath bins and concentrate on the central and outer magnetosheath properties.

The bottom panels of Figure 1 show the asymmetry (see Eq. 1) of the magnetic field strength in the central ($1/3 < F_{\text{Msheath}} < 2/3$) and outer ($2/3 < F_{\text{Msheath}} < 1$) magnetosheath as a function of the angle from the Sun-Earth line. The asymmetry level is

obtained from both a spatial average of this parameter inside each azimuthal bin and a temporal average over 150 s of the simulation, in order to minimise the effects of transient structures in the magnetosheath. The error bars associated with the asymmetry are very small, of the order of $0.1 - 0.2\%$, compared to those from spacecraft observations (e.g. Dimmock et al., 2017), most likely due to the steady upstream conditions in our runs and the large number of simulation cells in each spatial bin. Figures 1d and e reveal a definite $Q_\perp$-favoured asymmetry (positive values of the asymmetry) in all three runs. In Run 1, which corresponds to a typical Parker-spiral IMF orientation at Earth, we find an asymmetry level ranging between $0$ and $15\%$ in the central magnetosheath. The asymmetry level is significantly larger just downstream of the shock, suggesting that the field line draping and pile-up in front of the magnetosphere tend to smooth out the effects of the bow shock. Our results are in good agreement with the $0 - 10\%$ $Q_\perp$-favoured asymmetry obtained by Dimmock et al. (2017) based on statistics of spacecraft data. This $Q_\perp$-favoured asymmetry is due to the stronger compression of the magnetic field at the quasi-perpendicular bow shock, because only the tangential magnetic field components are enhanced at the bow shock crossing, while the normal component remains unchanged (Treumann, 2009; Hoilijoki et al., 2019).

When the cone angle is reduced from $45°$ to $30°$ in Runs 2A and 2B, the asymmetry becomes stronger in the central magnetosheath, exceeding 40% on the flanks in Run 2A. This is most likely due to the quasi-parallel sector of the shock being shifted closer to the subsolar point, and thus affecting a larger fraction of the dayside magnetosheath. Because of the magnetosheath flow pattern, the plasma entering the magnetosheath in the subsolar region then populates a large fraction of the flank magnetosheath. As a result, the regions of very low magnetic field strength (in dark blue in the bottom parts of panels a-c), due to the weak magnetic field compression at the quasi-parallel shock crossing, extend over most of the dawn side magnetosheath, forming a starker contrast with the dusk sector. We also note that they penetrate deeper in the magnetosheath, resulting in similar levels of magnetic field asymmetry in the outer and the central magnetosheath in Runs 2A and 2B. This contrast between Run 1 and Runs 2A and 2B may be related to the different draping pattern of the field lines at lower cone angle.

The magnetic field asymmetry is significantly weaker in Run 2B than in Run 2A. This lower asymmetry level at lower $M_A$ is most likely due to the reduced magnetic field compression affecting more strongly the magnetic field strength downstream of the quasi-perpendicular bow shock. To confirm this, we calculate the magnetic field strength just downstream of the bow shock based on the Rankine-Hugoniot jump conditions and assuming magnetic coplanarity is satisfied. We use the solar wind parameters of the Vlasiator runs as upstream conditions. The downstream to upstream ratio of the magnetic field magnitude is displayed in Figure 2 as a function of $\theta_{Bn}$ and $M_A$. This clearly shows that the magnetic field compression at the quasi-parallel bow shock does not vary with $M_A$ for the considered $M_A$ range, while higher values are reached on the quasi-perpendicular side as $M_A$ increases. These different behaviours on the quasi-parallel and the quasi-perpendicular sectors as a function of $M_A$ result in a less pronounced asymmetry at lower $M_A$.

Finally, we observe a gradual increase in the asymmetry from the subsolar region towards the flanks. This is likely due to the variation of the $\theta_{Bn}$ angle along the bow shock surface. Using a bow shock fit and the IMF direction, we estimated the value of $\theta_{Bn}$ along the bow shock. In Run 1, $\theta_{Bn}$ increases from the bow shock nose to the terminator on the quasi-perpendicular side, while it decreases at a similar rate on the quasi-parallel side, reaching its extrema on both flanks in the last azimuthal bin

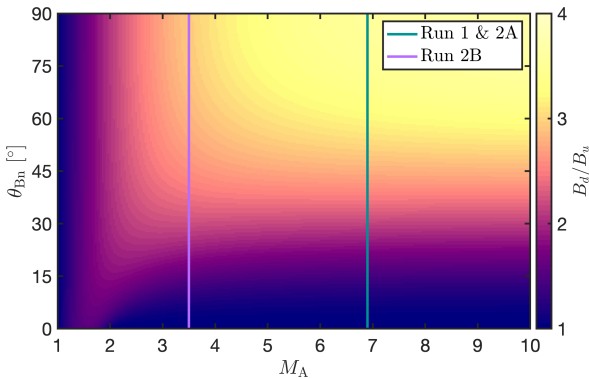

**Figure 2.** Downstream to upstream ratio of the magnetic field strength as a function of the Alfvén Mach number $M_A$ and the $\theta_{Bn}$ between the IMF direction and the shock normal, calculated based on the Rankine-Hugoniot relations.

near the terminator. We have $\theta_{Bn} \sim 0°$ ($\theta_{Bn} \sim 90°$) near the terminator on the quasi-parallel (quasi-perpendicular) flank. In Runs 2A and 2B, $\theta_{Bn}$ also increases with the azimuthal angle on the quasi-perpendicular side, but on the quasi-parallel sector, it first decreases until reaching 0 at around $45°$ from the Sun-Earth line, and then starts increasing again. The magnetic field asymmetry keeps increasing beyond this point probably because the asymmetry level is computed in a broad area and not just in the close vicinity of the bow shock, and other effects than shock compression come into play in the magnetosheath, for example field line pile-up and draping around the magnetosphere.

### 3.2 Ion bulk velocity

Figure 3 displays the plasma bulk velocity normalised to the solar wind speed in the three runs, and its associated asymmetry in the central and outer magnetosheath, in the same format as Figure 1. Again, the asymmetry is calculated based on a 150 s average of the bulk velocity inside each of the magnetosheath bins. As expected, the plasma velocity is very low in the subsolar magnetosheath, while the flow is faster on the flanks, because the tangential velocity is mostly preserved at the shock while its normal component is reduced, according to Rankine-Hugoniot relations.

Figure 3d shows a pronounced $Q_\perp$-favoured asymmetry in the central magnetosheath, with an asymmetry level ranging between 10 and 20% in Run 1 and in Run 2B. In Run 2A, very high values, over 25%, are reached in some azimuthal bins close to the subsolar region, but the overall asymmetry level appears only marginally higher than in the other runs. Dimmock and Nykyri (2013) and Dimmock et al. (2017) evidenced a $Q_\perp$-favoured asymmetry in their statistical data set, albeit with values somewhat below those found in our simulations, between 0 and 10%. Walsh et al. (2012) also reported a velocity asymmetry with the same polarity in spacecraft measurements and in MHD simulations.

In the outer magnetosheath, the level of the asymmetry tends to decrease when moving away from the subsolar region, except in the last two azimuthal bins in Run 1. As illustrated in Fig. 4, which shows the average velocity in the outer magnetosheath as a function of the angle from the Sun-Earth line, the flow speed increases more rapidly on the quasi-parallel flank than on the

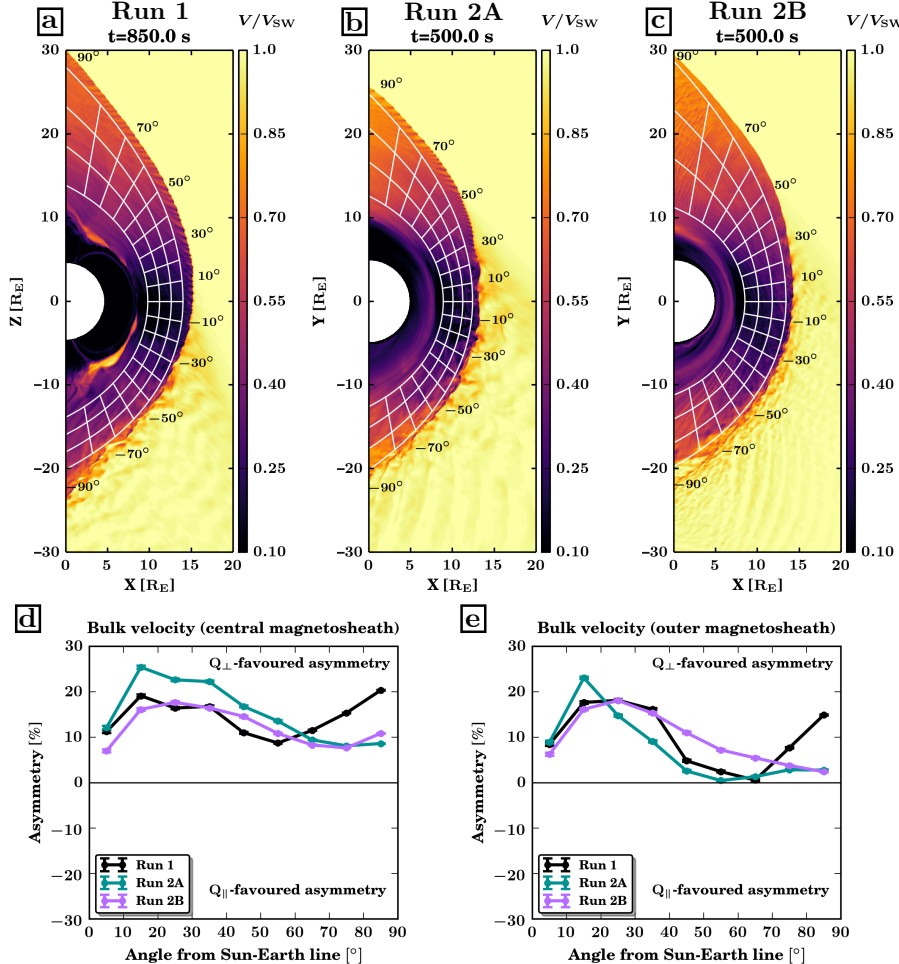

**Figure 3.** Top panels: ion bulk velocity in the simulation plane, normalised with the solar wind speed, in Run 1 at time $t = 850\,\text{s}$ (a), in Run 2A (b) and 2B (c) at time $t = 500$ s. The spatial bins used to calculate the average magnetosheath parameters are shown in black. Bottom panels: magnetic field strength asymmetry in the central (d) and outer (e) magnetosheath.

quasi-perpendicular flank. This progressively smoothes out the difference between both sectors. Also, the fact that the velocity
is larger further down on the flanks tends to reduce the asymmetry level, as the same absolute difference in velocity between the quasi-parallel and quasi-perpendicular sectors results in a smaller value of the asymmetry, which is calculated as the relative difference (see Eq. 1).

Beyond $40°$ from the Sun-Earth line, the asymmetry level reduces to values close to 0 in Run 2A and partly in Run 1. Only in Run 2B does the asymmetry remain persistently $Q_\perp$-favoured across the entire dayside magnetosheath. The re-increase of
320 the asymmetry level in the last two azimuthal bins in Run 1 reflects an abrupt decrease in velocity near the terminator on the

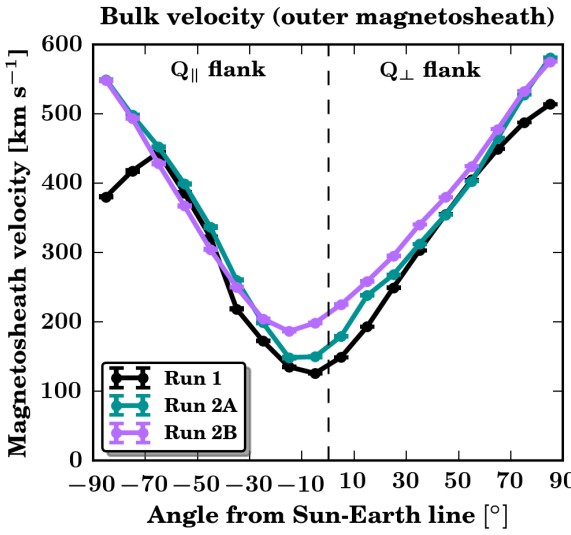

**Figure 4.** Bulk velocity in the outer magnetosheath as a function of the angle from the Sun-Earth line in all three runs.

quasi-parallel flank. This likely stems from the irregular shape of the bow shock in Run 1, which bulges outward beyond $-70°$ from the Sun-Earth line due to a large and persistent foreshock transient.

Figure 5 displays the shock density compression ratio as a function of $\theta_{Bn}$ for the two different $M_A$ values in Runs 1 and 2A ($M_A = 6.9$) and in Run 2B ($M_A = 3.5$). As illustrated in Fig. 5, the density compression ratio is roughly constant over the whole $\theta_{Bn}$ range for the $M_A$ of Runs 1 and 2A (in green), while it is considerably lower on the quasi-perpendicular side than on the quasi-parallel side at the lower $M_A$ of Run 2B (in purple). This could explain why the velocity asymmetry level is essentially larger in Run 2B than in Run 2A in the outer magnetosheath (Fig. 3e). This trend however disappears deeper in the magnetosheath (Fig. 3d), probably because other processes affect there the magnetosheath flow.

Our simulations also show that the flow stagnation region is slightly shifted from the subsolar point towards the quasi-parallel magnetosheath (see Fig. 4 where the dashed line indicates the subsolar point). In Run 1, the velocity minimises at about $10°$ from the Sun-Earth line on the quasi-parallel side. This is probably due to the velocity deflection at the bow shock which depends on $\theta_{Bn}$, as predicted by the Rankine-Hugoniot jump conditions to preserve the continuity of the tangential electric field (e.g., Treumann, 2009). As a result, asymmetric flow speeds are observed when comparing the quasi-perpendicular and quasi-parallel magnetosheath. Field line draping around the magnetosphere may also play a role in reducing the velocity in the quasi-parallel magnetosheath. The shift of the stagnation region towards the quasi-parallel flank is slightly greater for a $30°$ cone angle (Runs 2A and 2B), consistent with the $\theta_{Bn}$ dependence of the velocity deviation at the bow shock.

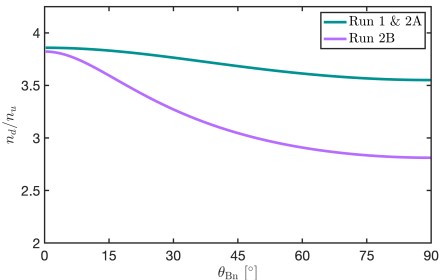

**Figure 5.** Density compression ratio as a function of $\theta_{\mathrm{Bn}}$ for two different $\mathrm{M_A}$ values, corresponding to those in the simulation runs.

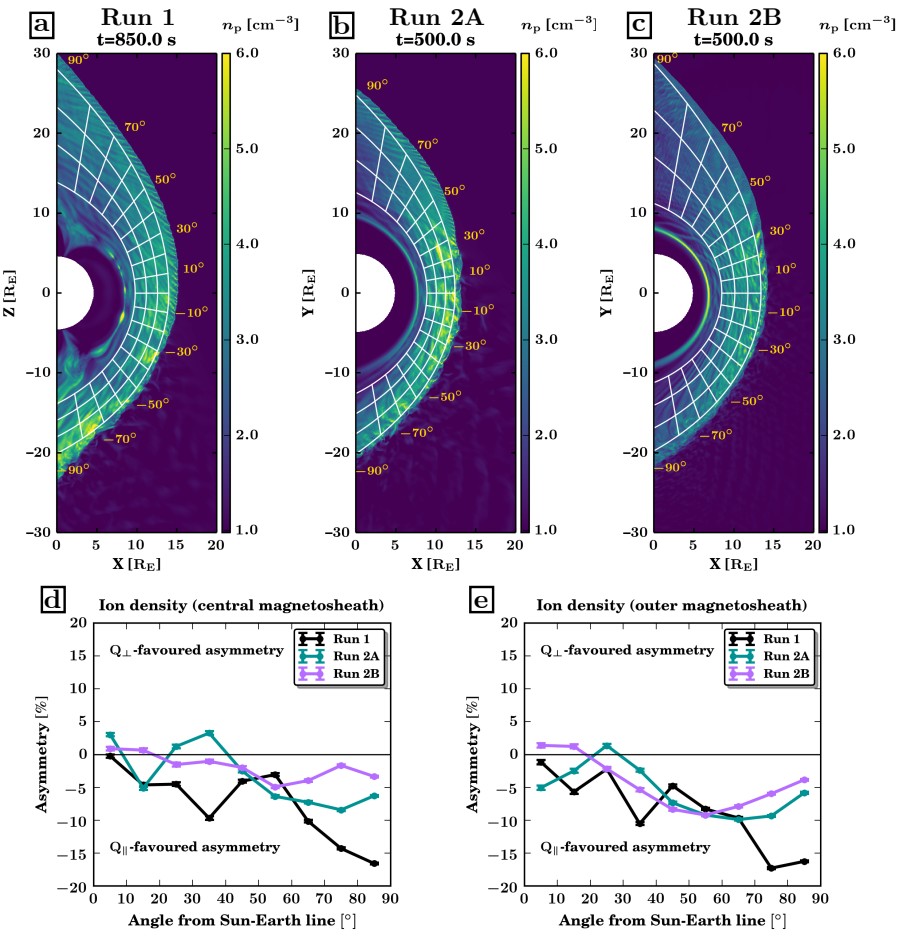

**Figure 6.** Same format as in Figure 3 but for the ion density.

### 3.3 Ion density

Plotted in Fig. 6 is the ion density and its associated asymmetry in the central and outer magnetosheath, in the same format as Figures 1 and 3. The upper panels show that the ion density in the magnetosheath is essentially up to four times its upstream value, consistent with previous works and with the theoretical density compression ratio at the bow shock (Formisano et al., 1973). A few regions of larger density enhancements (in yellow) are observed downstream of the quasi-parallel shock. Similar transient density enhancements are seen throughout the 150 s of simulated time which are used to calculate the magnetosheath asymmetry. Such large densities in the magnetosheath, exceeding the theoretical MHD limit, are a common feature in hybrid-kinetic simulations of the bow shock/magnetosheath system (see for example Omidi et al., 2014; Karimabadi et al., 2014). They are probably due to density enhancements in the foreshock which are advected and compressed through the bow shock, and appear to be associated with ripples of the shock front.

Figures 6d and 6e evidence a mostly $Q_\parallel$-favoured asymmetry of the ion density in the magnetosheath. However, in Runs 2A and 2B, associated with a $30°$ cone angle, multiple azimuthal bins near the subsolar region display an opposite polarity of the asymmetry, both in the outer and the central magnetosheath. Moreover, we note that the values of the density asymmetry are much more sensitive to the time interval over which the data are averaged than for the other parameters under study. This is probably due to the variability of the plasma density just downstream of the quasi-parallel shock. The patches of high density alternate with depleted regions, which result in $Q_\perp$-favoured asymmetries in some azimuthal bins, even when performing 150 s temporal averages. This demonstrates the high variability of the magnetosheath density, even under completely steady solar wind conditions. For example, in Run 2A, we note that patches of high density just downstream of the bow shock are concentrated in the subsolar magnetosheath and are distributed on either sides of the Sun-Earth line, as evidenced in Figure 6b. This could explain the reversed polarity of the asymmetry in some azimuthal bins near the subsolar point.

The asymmetry levels appear to be essentially similar when comparing the different runs. The $Q_\parallel$-favoured asymmetry might be more pronounced near the terminator in Run 1 than in the other runs, but the large fluctuations of the asymmetry level from one bin to another makes it difficult to ascertain. As mentioned in Section 3.2, the shock compression ratio shows little dependence on $\theta_{Bn}$ in the range of $M_A$ associated with Runs 1 and 2A, while it is significantly lower on the quasi-perpendicular flank than on the quasi-parallel flank in the low $M_A$ range, such as in Run 2B. Therefore, according to the MHD theory, the density asymmetry should be stronger at lower $M_A$. We do not observe however a significant variation of the asymmetry level between Runs 2A and 2B, possibly due to the spatial variability of the magnetosheath density, or to the low cone angle value. The flatter shape of the bow shock at lower $M_A$ would also tend to counteract the effect of the density compression ratio, as only a smaller range of $\theta_{Bn}$ values is found at the surface of a more planar bow shock.

Finally, we note that the variability of the density in the outer magnetosheath is much lower at reduced $M_A$, which results in a smoother distribution of the asymmetry. This could be related to foreshock disturbances being weaker at lower $M_A$, since the density of suprathermal ions is reduced (Turc et al., 2015, 2018).

### 3.4 Comparison with spacecraft observations

We now compare our numerical results with the asymmetries obtained from a statistical data set of magnetosheath observations from the Time History of Events and Macroscale Interactions during Substorms (THEMIS) spacecraft (Angelopoulos, 2008; Dimmock and Nykyri, 2013; Dimmock et al., 2017). The data were collected between January 2008 and December 2017 and are binned according to the spacecraft coordinates in the Magnetosheath InterPlanetary Medium (MIPM) reference frame (Bieber and Stone, 1979; Dimmock et al., 2017). In this coordinate system, the $x-$axis points opposite to the solar wind flow, while the $y-$axis is defined such that the quasi-perpendicular sector of the bow shock lies in the $+y$ direction and its quasi-parallel sector at negative $y$. This ensures that all data associated with a given shock regime are grouped together on one side of the magnetosheath. The $z-$axis completes the orthogonal set. Then the radial coordinate of each measurement point is calculated as the fractional distance between a model bow shock and magnetopause, which removes the effects of the motion of these boundaries due to changing upstream conditions. The data points are thus organised with their fractional distance inside a normalised magnetosheath and with their azimuthal angle from the Sun-Earth line. Each measurement point is associated with a set of upstream conditions, based on the OMNI data (King and Papitashvili, 2005) at the time of the THEMIS observations. More details on the data processing can be found in Dimmock and Nykyri (2013); Dimmock et al. (2017) and the references therein.

As in previous studies using this statistical data set (e.g. Dimmock et al., 2015a; Dimmock et al., 2017), we concentrate only on measurements in the central magnetosheath, that is, where $1/3 < F_{\mathrm{Msheath}} < 2/3$, to avoid including data from other regions in case of inaccuracies in the determination of the boundary position. The average parameters in the central magnetosheath are computed inside $15°$-wide angular bins, with a $50\%$ overlap between two consecutive bins. The asymmetry is then calculated using Eq. 1. Furthermore, we divide the statistical data set into two ranges of cone angles, depending on the IMF orientation associated with each of the magnetosheath measurements. The magnetosheath asymmetries associated with a cone angle close to that of the Parker spiral orientation ($40° < \theta_{\mathrm{Bx}} < 50°$) are shown in black in Figure 7 and those associated with a lower cone angle value ($20° < \theta_{\mathrm{Bx}} < 35°$) are plotted in blue. We note here that the data set contained too few data points at $\mathrm{M_A} < 5$ for us to investigate the change in the asymmetries at low Alfvén Mach number.

Firstly, we find an excellent agreement between simulations and observations regarding the polarity of the asymmetry for the three parameters considered here, as noted already in the previous sections. The levels of asymmetry tend however to be lower in the observational data compared to the simulations. This could be due to the processing method of the statistical data set, which calculates averages over very diverse upstream conditions, and thus results in a conservative estimate of the asymmetry.

As concerns the influence of the cone angle, the statistical data do not show evidence of a significant increase in the magnetic field strength asymmetry when the cone angle is reduced, contrary to our numerical simulations. The density asymmetry displays much more spatial variability at low cone angle, with about half of the azimuthal bins having a $\mathrm{Q_\perp}$-favoured asymmetry, while most of them showed a clear $\mathrm{Q_\parallel}$-favoured asymmetry for a Parker spiral IMF orientation. This agrees well with the numerical results presented above, and is likely due to foreshock processes causing enhanced variability of the magnetosheath density at lower cone angles.

## 4 Discussion

We have quantified the asymmetry of the magnetic field magnitude, ion density and bulk flow velocity inside the dayside magnetosheath in three Vlasiator global runs with different IMF conditions. We note that the use of global ion-kinetic simulations presents several main advantages.

First, the global coverage of the magnetosheath for a given set of solar wind conditions provided by the simulations allows us to investigate the asymmmetries both in the central and the outer magnetosheath. In contrast, observational studies are often restricted to the central magnetosheath to make sure that the data set does not include magnetosphere or solar wind measurements (e.g. Dimmock et al., 2015a; Dimmock et al., 2017), or to locations just outside the magnetopause to avoid relying on boundary models to estimate the position inside the magnetosheath (Walsh et al., 2012). The comparison of the asymmetry levels in the central and outer magnetosheath provides us with new information regarding the influence of the bow shock on the magnetosheath parameters, in particular on the magnetic field asymmetry, which is stronger just downstream of the shock than deeper in the magnetosheath.

Second, the simulations enable us to investigate the asymmetry levels at low Alfvén Mach number ($M_A \sim 3.5$, Run 2B), while the statistical data set compiled from THEMIS measurements does not contain enough data points at such low $M_A$ to derive the asymmetry of the magnetosheath parameters. This is why we did not compare our numerical results concerning $M_A$ with observations in Section 3.4. Low Alfvén Mach numbers are encountered only occasionally at Earth, but they are of great importance for solar wind-magnetosphere coupling because they are associated with extreme solar wind disturbances such as magnetic clouds (Turc et al., 2016) and they result in atypical conditions in the magnetosheath (Lavraud and Borovsky, 2008; Lavraud et al., 2013). Other studies have suggested that the Alfvén Mach number plays a role in the asymmetry (Walsh et al., 2012; Dimmock et al., 2017) but it is difficult to make a direct and meaningful comparison between all of these studies since there are extensive differences across methodologies, models, and datasets. However, there are clearly unanswered questions which deserve further study and may be addressed with future missions and/or model runs.

Third, the inclusion of ion kinetic physics in the simulations makes it possible to study the effects of the quasi-parallel shock and its associated foreshock on magnetosheath parameters. These effects are particularly substantial for the ion density, whose variability in the magnetosheath is driven by quasi-parallel bow shock and foreshock processes. The alternating patches of higher and lower densities, which are chiefly responsible for the varying asymmetry levels in the outer magnetosheath, appear to be associated with irregularities of the shock front, whose scale is comparable to that of the foreshock waves, consistent with previous studies which have established that foreshock waves modulate the shape of the shock front (e.g. Burgess, 1995).

The main limitation of our numerical simulations is the 2D set-up, which results in particular in enhanced field line pile-up in front of the magnetopause, and thus causes a slow outward motion of the bow shock. Therefore, the magnetosheath thickness is somewhat overestimated in the later times of our runs. However, this should not alter the global magnetosheath parameters, except near the magnetopause where the pile-up takes place. We verified that this does not affect significantly the asymmetry levels, and found that the temporal variability of the asymmetries in the simulation was caused by transient processes rather than by the shock progressive expansion. The 2D set-up may also influence the field line draping pattern in the magnetosheath,

which may affect the extent of the region of low magnetic field strength observed in the central magnetosheath downstream of the quasi-parallel shock when the cone angle is reduced to $30°$. In contrast, the magnetic field strength is higher in the central magnetosheath than in the outer magnetosheath for a $45°$ cone angle in Run 1 (see Figure 1a and 1b). Future 3D simulations could allow us to evaluate if the asymmetry is less pronounced in this region than in the outer magnetosheath when field lines can flow around the magnetosphere.

Another possible limitation of our simulations is the spatial resolution, which corresponds to $1.3$ solar wind ion inertial lengths in Run 1 and $1$ ion inertial length in Runs 2A and 2B. As a result, waves with a wavelength below this spatial resolution are not included in our simulations. This resolution is however sufficient to resolve the dominant low-frequency wave modes in the magnetosheath, namely the mirror and the Alfvén ion cyclotron waves (Hoilijoki et al., 2016; Dubart et al., 2020). At the shock front, a cell size of $1$ ion inertial length or larger may not correctly evaluate the gradient in the ramp. However, the hybrid-Vlasov formalism based on distribution functions enables the use of a slope limiter which allows for total variation diminishing evolution of discontinuities and steep slopes even at somewhat lower resolution. The shock transition is therefore well described in our simulations, and the downstream parameters are correctly modelled. The study we present here focuses on the large-scale distribution of magnetosheath properties. Therefore, the spatial resolution in our Vlasiator runs is sufficient to study global magnetosheath parameters and how they are impacted by ion kinetic physics.

We note that the levels of asymmetry obtained from the numerical simulations are larger than those from the observational data set, for all parameters considered in this study. This is probably due to the fundamentally different methods through which the magnetosheath parameters were obtained. In the simulations, the asymmetry is calculated based on spatial averages of the magnetosheath parameters for a single set of steady upstream conditions, while observational results are a compilation of localised measurements taken during a variety of upstream conditions. Specifically, the IMF can assume any orientation in the observational data set, including in particular an out-of-plane component while the THEMIS spacecraft orbit near the Earth's equatorial plane. Even though the MIPM reference frame arranges the measurements corresponding to the quasi-parallel/quasi-perpendicular sectors on the negative/positive $y-$hemispheres, it does not account for the different cone angles nor for the out-of-plane IMF component. As a result, data points associated with widely different $\theta_{\mathrm{Bn}}$ values can be grouped together. Also, some data points may be misidentified as quasi-parallel or quasi-perpendicular because the upstream conditions are determined from the OMNI propagated data set which may not reflect exactly the actual conditions at Earth's bow shock. These two effects would tend to smooth out the asymmetries in the statistical data set. The numerical simulations, on the other hand, do not suffer from these limitations, resulting in more pronounced asymmetries. A similar interpretation was proposed by Walsh et al. (2012), who also found larger asymmetry levels in their MHD simulations than in the observations. This further supports that the apparent discrepancy between observations and simulations is only a natural consequence of the different methods used for obtaining the average magnetosheath parameters.

The magnetic field asymmetry also behaves differently in the observations and the simulations when changing the cone angle. In Vlasiator, we find a significant increase of the asymmetry at low cone angle, whereas no significant variation is observed in the statistical THEMIS data set. It should be noted that the spacecraft observations are not associated with a single value of the IMF cone angle, but are a compilation of measurements taken for a range of cone angles, between 20 and $35°$.

As the IMF becomes more radial, the quasi-parallel sector of the bow shock and its associated foreshock move closer to the subsolar point. For a purely radial IMF, the magnetosheath asymmetries due to the bow shock configuration should completely disappear, as the $\theta_{\mathrm{Bn}}$ values are then distributed symmetrically about the Sun-Earth line (see e.g. Turc et al., 2016). Therefore, there should be a value of the cone angle at which the magnetosheath asymmetries maximise, before decreasing when further reducing the cone angle to finally reach the symmetrical configuration for a purely radial IMF. The range of cone angles used in collating the statistical data might therefore contain significant variation in asymmetry levels. This in turn could explain why the asymmetry level for $20 - 35°$ cone angles remains the same as for $40 - 50°$ cone angles in the observations.

Using a semi-empirical model of the magnetosheath magnetic field (Turc et al., 2014), we calculate the asymmetry level of the magnetic field strength associated with the same upstream parameters as in Run 1 and Run 2A. The model predicts a higher asymmetry level at $30°$ than at $45°$ cone angle (not shown), in agreement with our numerical simulations. This lends further support to the hypothesis that the different behaviour in spacecraft measurements could be due to the array of solar wind conditions and IMF orientations included in the statistical data set. Also, the data could be affected by processes at smaller spatial scales than those resolved in our simulations, though it is unlikely that this will play a significant role here, since the data are averaged over several minutes.

The ion density asymmetry was essentially $Q_{\parallel}$-favoured in all our runs, consistent with previous observational and numerical works (Paularena et al., 2001; Longmore et al., 2005; Walsh et al., 2012; Dimmock et al., 2016b) and MHD theory (Walters, 1964). It should be noted however that the most recent studies by Dimmock et al. (2016b) and Dimmock et al. (2017) only found a clear $Q_{\parallel}$-favoured asymmetry near the magnetopause, while no clear polarity was observed in the central magnetosheath. In our simulations, we found in several instances that the asymmetry in some of the azimuthal bins displayed an opposite polarity. We also observed a large temporal variability of both its level and its polarity in our simulations, despite the completely steady upstream conditions. This suggests that the magnetosheath density is extremely sensitive to transient processes, originating for example in the foreshock and at the quasi-parallel bow shock. The fluctuations that are typically present in the solar wind parameters would be conducive to even more variability of the magnetosheath density. The inconclusive results regarding the polarity of this asymmetry in the central magnetosheath (Dimmock et al., 2016b; Dimmock et al., 2017) and the large discrepancies in the asymmetry levels quantified in various studies (see the summary table in Walsh et al., 2014) likely stem from this high variability.

## 5 Conclusions

In this work, we studied the asymmetry between the quasi-parallel and the quasi-perpendicular sectors of the Earth's magnetosheath using global hybrid-Vlasov simulations. We quantified the level of asymmetry in the central and outer magnetosheath for the magnetic field strength, ion density and bulk velocity and investigated its variation when reducing the cone angle and the $M_A$. For all parameters, we find a polarity of the asymmetry ($Q_\perp$-favoured or $Q_\parallel$-favoured) that is consistent with earlier works (see Dimmock et al., 2017, for a recent review). The asymmetry levels tend to be higher in the numerical simulations, due to the fact that the magnetosheath parameters are obtained for a given set of fixed upstream conditions in the model, in-

stead of a compilation of normalised localised measurements. Using a set of three runs with different upstream conditions, we investigated for the first time how the asymmetries change when the angle between the IMF and the Sun-Earth line is reduced and when the Alfvén Mach number decreases.

For a $30°$ cone angle, we found similar levels of magnetic field asymmetry in the outer and central magnetosheath, while they differed significantly at a larger cone angle. We also noted that the polarity of the density asymmetry reversed in some bins near the subsolar region, likely due to the quasi-parallel sector of the bow shock being located closer to the subsolar point. The magnetic field strength asymmetry increased significantly at $30°$ cone angle, possibly due to the low $\theta_{\mathrm{Bn}}$ near the bow shock nose resulting in a reduced magnetic field compression across most of the quasi-parallel flank of the magnetosheath. This effect was however not observed in the statistical data sets obtained from spacecraft measurements.

Reducing the $\mathrm{M_A}$ results in a less pronounced magnetic field asymmetry because of the weaker compression of the magnetic field at the quasi-perpendicular bow shock, while that at the quasi-parallel shock remains roughly unchanged. We also noted that the density asymmetry displays less variability, probably due to weaker foreshock and quasi-parallel shock disturbances at lower $\mathrm{M_A}$. This change is particularly visible here because of the low cone angle, but may be less discernable for less radial IMF orientations, as the foreshock will retreat towards the flank. Future simulation runs with a low $\mathrm{M_A}$ and a larger cone angle could allow to test this.

It is worth noting that even for completely steady upstream conditions, the magnetosheath density shows significant temporal and spatial variations, in particular downstream of the quasi-parallel shock. These variations are likely caused by foreshock and quasi-parallel shock transient processes. They can influence noticeably the level of asymmetry in some parts of the magnetosheath, and even cause reversals of its polarity in some azimuthal sectors. Our results show that density asymmetry variations in the magnetosheath are an inherent effect of the bow shock and foreshock, instead of a statistical artefact. This is most likely one of the sources for the wide variety of levels of density asymmetry quantified in previous observational studies.

This work shows that global kinetic simulations provide a reliable tool to study magnetosheath asymmetries. The global coverage of the magnetosheath obtained in each run allows for a precise quantification of the asymmetry levels for a given set of solar wind conditions, in contrast with spacecraft statistical data sets which quantify the average value of the asymmetries across a wide range of upstream conditions. Moreover, the inclusion of ion kinetic physics is necessary to properly describe the dynamics of the quasi-parallel shock which affect strongly the variability of the magnetosheath density. Numerical simulations also enable us to perform parametric studies, thus allowing us to study the influence of specific upstream parameters. Here we limited our analysis to three runs because of the large computational cost of Vlasiator simulations, but future studies could make use of larger sets of runs, with more varied upstream conditions, once they become available.

*Code availability.* Vlasiator (http://www.helsinki.fi/en/researchgroups/vlasiator/, Palmroth, 2020) is distributed under the GPL-2 open source license at http://github.com/fmihpc/vlasiator/ (Palmroth and the Vlasiator team, 2020). Vlasiator uses a data structure developed in-house (https://github.com/fmihpc/vlsv/, Sandroos, 2019), which is compatible with the VisIt visualization software (Childs et al., 2012) using a plugin available at the VLSV repository. The Analysator software (https://github.com/fmihpc/analysator/, Hannuksela and the Vlasiator team,

2020) was used to produce the presented figures. The runs described here take several terabytes of disk space and are kept in storage maintained within the CSC – IT Center for Science. Data presented in this paper can be accessed by following the data policy on the Vlasiator web site.

*Author contributions.* L.T. initiated and coordinated the study, performed part of the data analysis and wrote the first draft of the manuscript. V.T. developed the methodology and performed the initial data analysis. A.D. provided the THEMIS statistical data set and helped in the interpretation of the results. A.J. provided Figures 2 and 5, and contributed to the comparison with the Rankine-Hugoniot jump conditions. M.P. is the PI of the Vlasiator model and gave inputs to the interpretation of the simulation results. M.B., U.G. and Y.P.-K. ran the Vlasiator runs used in this study. Together with A.J., M.G. and M.D., they contributed to the analysis of the Vlasiator outputs. All co-authors participated in the discussion of the results and contributed to improving the manuscript.

*Competing interests.* The authors declare that they have no conflict of interest.

*Acknowledgements.* This project has received funding from the European Union's Horizon 2020 research and innovation programme under the Marie Sklodowska-Curie grant agreement No 704681. The work of L.T. is supported by the Academy of Finland (grant number 322544). We acknowledge the European Research Council for Starting grant 200141-QuESpace, with which Vlasiator was developed, and Consolidator grant 682068-PRESTISSIMO awarded to further develop Vlasiator and use it for scientific investigations. The work leading to these results has been carried out in the Finnish Centre of Excellence in Research of Sustainable Space (Academy of Finland grant numbers 312351 and 312390). V.T. acknowledges the Academy of Finland (grant number 328893). A.P.D. was funded by the Swedish Civil Contingencies Agency grant 2016-2102. We acknowledge PRACE Tier-0 grant number 2016153521 with which Run 1 was carried out in CINECA, Italy. The CSC - IT Center for Science in Finland is acknowledged for the Grand Challenge awards leading to the results shown here for Runs 2A and 2B.

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

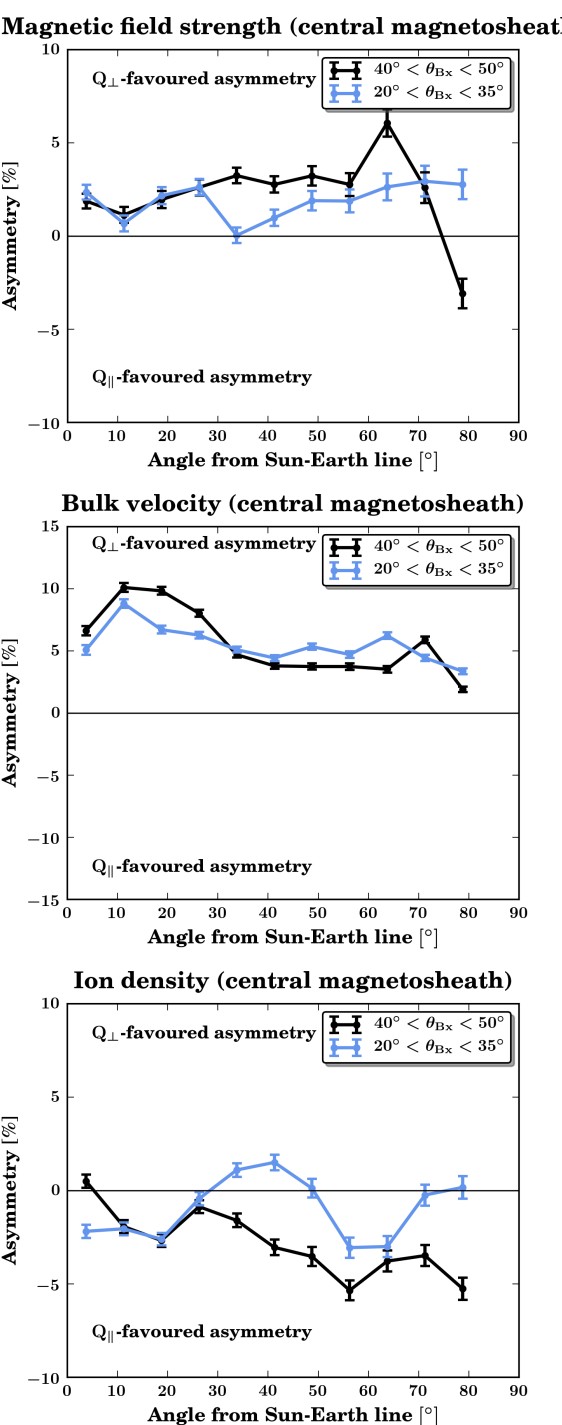

**Figure 7.** Asymmetries in the central magnetosheath as obtained from statistics of THEMIS spacecraft observations. From top to bottom: magnetic field strength, bulk velocity and ion density. The black curves correspond to data with a cone angle near the Parker spiral orientation ($40° < \theta_{\mathrm{Bx}} < 50°$) and the blue curves to data with a low cone angle values ($20° < \theta_{\mathrm{Bx}} < 35°$).