# Peer review of "Asymmetries in the Earth's dayside magnetosheath: results from global hybrid-Vlasov simulations"

_Annales Geophysicae, 2020_

## Referee Comment (RC1) · Anonymous Referee #1 · 18 Apr 2020

Summary of the manuscript: Authors have studied magnetosheath dawn-dusk asymmetries of the magnetic field (B), proton number density (n) and plasma flow speed (V) in the 2-D global Vlasiator simulation and compared the results with statistics from the THEMIS observations. The main conclusion made by the authors is that while the polarity of the asymmetries agrees with the THEMIS results, the magnitudes are in disagreement because Vlasiator is run with one set of conditions, whereas spacecraft observations show cumulation of the observations from different solar wind conditions and IMF orientations.

Overall evaluation of the manuscript:

Unfortunately, in its present form I am unable to recommend this manuscript for publication in the scientific literature for the following reasons:

1) There are no original ideas or new scientifically valid results. There have been several past studies of the magnetosheath properties using both spacecraft data and different plasma approximation (e.g., MHD, hybrid, kinetic-simulations). Some of these have been cited but not all: The bow shock is not the only source for magnetosheath plasma. Also magnetopause processes are important that can transport magnetospheric particles into the magnetosheath. The study of the dawn-dusk asymmetries of B, n and V using spacecraft data has already been done. If the motivation is to solely test the code robustness, by using the study of dawn-dusk asymmetries as a validation effort against spacecraft data, a technical paper may be more suitable.

2) For scientific paper "more in depth" analysis of the physical mechanisms is required to address what differences are due to numerical issues, what are due to kinetic physics, and what are the mechanisms. For example, how are the magnetosheath densities > 4 explained, if solar wind density is 1? How are these results affected by grid-resolution.

3) The Methodology of the present paper is flawed so no accurate scientific conclusions can be made at this time. The authors' conclusion, that the disagreement with the data and simulation is due to the fact that simulation is run for one set of conditions, whereas spacecraft data contains the history of IMF and solar wind, is only one possible reason. Paper fails to discuss the other, more plausible and likely more significant reasons listed below:

i) Code is run with 750 km/s solar wind speed. This condition occurs rarely as the average solar wind speed is slightly less than 400 km/s. Therefore, it is not logical or scientifically justified to compare the runs with 750 km/s solar wind velocity with the spacecraft statistics collected in the magnetosheath, when the solar wind flow preceding the THEMIS data collection is about 400 km/s. With higher solar wind speeds,

the shock compression becomes stronger and one would expect higher magnetic field strengths downstream of the quasi-perpendicular shock than for solar wind speeds of 400 km/s.

ii) The runs use upstream solar wind density of 1/cc, which would result in a maximum downstream density of about 4/cc downstream of quasi-perp shock. Assuming the values of 1- 4/cc, the ion inertial length would be 228 km to 114 km in the magnetosheath, respectively, and for higher densities these scales get even smaller. The paper does not describe what the coordinate-space resolution is used for the runs. In order to appropriately resolve the physical ion scales, the Vlasiator should use about 5 to 10 cells/ion inertial length, which requires coordinate space resolution at the minimum of about 20-40 km, otherwise the kinetic effects can artificially dominate at larger length scales than in the real system. Furthermore, if the ion inertial scale at the bow shock is not appropriately resolved, the results pertaining to kinetic shock physics are physically meaningless or exaggerated.

iii) Since the B, n and V are MHD quantities, authors should also run their case exactly with same parameters using the major global, state-of-the-art 3-D MHD codes available through Community Coordinated Modeling Center and compare their Vlasiator results with these.

Recommendation:

While I cannot recommend this paper for publication at this time, I hope the authors will use this feedback as an opportunity to improve the paper, and for transparency include additional missing information (e.g., the grid-resolution etc.). This work requires significant further code validation efforts in a global scale, where the effects of spatial grid-resolution are systematically studied, so that the results can be correctly interpreted.

Please see the specific comments below that need to be addressed after which the paper may eventually become suitable for scientific literature:

Specific comments:

1. Calculate the statistical solar wind and IMF condition from THEMIS statistics used in the paper.

2. Plot and show a spatial map in x-y -plane for each run of the i) ion inertial length, ii) ion gyro-radius and iii) plasma beta, and collect a mean, minimum and maximum values of these at the central magnetosheath where the statistics pertaining to study is being collected during the course of the simulation.

3. Re-run Vlasiator with the statistical conditions and with the appropriate resolution (whichever length-scale is the smallest). For plasma beta of 1, the both length-scales should be the same. Compare the results with those in the present manuscript.

4. Add details and benchmarking how the phase space velocity distributions are processed to calculate n and V in 2-D-plane (as one can cut a 3-D velocity distribution in infinite ways). Show how the processing of the velocity distribution functions affects the results. Convince the reader of the validity of this processing at different regions in the magnetosheath.

5. Re-run Vlasiator with a) old-set of parameters shown in this manuscript while appropriately resolving these length-scales and compare with the results from original resolution.

6. Ion and electron temperatures are an important quantity to demonstrate the dawn-dusk asymmetry. It will be very interesting to see these two parameters, and the perpendicular and parallel temperatures.

7. Since the majority of the quantities compared in this study can also be obtained by the MHD global simulation, I suggest the authors also run the same solar input in the CCMC to compare with all the other three major MHD models and show the advantage of Vlasiator results.

8. What is the "zebra stripes" structure on the dusk side, are those physical waves

(which wave-mode) or a grid oscillation?

9. From MHD the maximum shock compression ratio would give magnetosheath densities of 4/cc if the density in the solar wind is 1/cc. Here the maximum density in the magnetosheath is 6/cc. Is this a kinetic effect and what is the physical mechanism to generate that? How is the area of the > 4/cc density regions in the magnetosheath dependent on the ion gyro-radius/inertial scale and grid resolution when compared to MHD simulations that are run with the same parameters and same resolution?

Minor comments:

1. The Discussion and Conclusions are repetitive. This could be made more concise.

2. The referencing is inadequate. Authors should extend their citations to include some of the following. Please see previous global hybrid simulation studies of the magnetosheath (e.g. by Y. Lin et al. (2001-2020), H. Karimabadi et al., N. Omidi et al,) and several missing studies related to spacecraft observations and statistics of the various fore-shock transient that modify magnetosheath properties (e.g., F. Plaschke et al. (2013-2019), D. Turner at al., H. Hietala et al.(2009-2018), T. Liu et al.(2017-2019), H. Zhang et al. (2013-,) as well as leakage of magnetospheric particles into the magnetosheath by various processes (e.g., I. Cohen et al. 2017; K. Sorathia et al., 2019), and due to local magentosheath physics (e.g., P. Gary et al., 2006 A. Retino et al. 2007, D. Sundkvist et al. 2007, J. Soucek et al.(2008-2015), V. Genot et al. (2001-2009), T. Phan et al., 2018).
* * *

---

## Referee Comment (RC2) · Anonymous Referee #2 · 26 Apr 2020

Referee report on the "Asymmetries in the Earth's dayside magnetosheath: results from global hybrid-Vlasov simulations", by L. Turc et al.

This manuscript describes 2D simulations of the dayside magnetosheath using the hybrid Vlasiator code, for three different upstream conditions (one in the noon-midnight plane, and two in the GSE equatorial plane). The authors appropriately describe the capabilities as well as the issues and shortcomings pertaining to this hybrid model. The detailed description of the challenges of magnetosheath studies using spacecraft observations is also highly appropriate. The explanations provided regarding the numerical results of magnetosheath asymmetries of parameters (B, density, and velocity)

[Figure]

downstream of the Qpara and Qperp bow shock regions as a function of angle from the Sun-Earth line are plausible, though perhaps not the only possible explanations. Comparing the numerical simulation results with magnetosheath observations by the THEMIS spacecraft is also highly appropriate.

There are two significant concerns with the manner in which the study results are presented. These ought to be fairly easily addressed, but are important because they directly affect most of the figures and results presented in this study:

1) Magnetosheath parameters determined from the numerical simulations within each spatial bin and for the time interval used are presented as averages; whereas the magnetosheath parameters determined from spacecraft observations are presented as medians. In order to ensure that the comparisons between simulations and observations are meaningful, the same statistical measure should be used for both (ideally medians, to avoid outlier kinetic effects due to processes at the bow shock convected into specific magnetosheath bins from unduly influencing the overall average value). An alternative is to demonstrate that within the magnetosheath bins, the distribution of values used to determine the spatial and temporal average is Gaussian, so that the average and median values are the same.

2) It is difficult to judge the robustness of the results, because there are no estimates of the statistical spread (uncertainties) associated with the averages (or medians). From the simulations, sampling in appropriately sized sub-spatial and sub-temporal bins to provide e.g., standard deviations (or quartiles) used in the estimate of the asymmetry would instill considerable confidence that the percentage of asymmetry results are robust. Similarly for the THEMIS observations, it would be more appropriate if statistical estimates representing the range of values within each bin are determined and then used to estimate the range of values (measure of uncertainty) for the percentages of asymmetry for the various plasma parameters.

Minor issues:

[Figure]

Line 268: considerable -> considerably

Figure 4: Should label which side of the plot corresponds to Qpara, and which side corresponds to Qperp.

Line 341: magnetosheah -> magnetosheath
* * *

---

## Referee Comment (RC3) · Anonymous Referee #3 · 28 Apr 2020

The paper describes the Earth magnetosheath response to the solar wind inflow using the Vlasiator code. The focus is put on the various asymmetries of plasma and magnetic parameters in three cases with varying IMF orientation and Alfven Mach number. The results are then compared to an analysis of THEMIS observations which was published previously (Dimmock et al.'s papers). The objectives are sound, the code and the analysis appropriate, however a number of key points make the paper not mature enough in the present form. They are listed first, then minor issues follow.

Major points:

- References: the references to previous works are not adequate. Concerning hybrid

codes for the magnetosheath, the literature was already vast before Vlasiator and 6D simulations of solar wind / planetary plasma interactions exist, e.g. Travnicek et al., 2007 (GRL), Hercik et al., 2013 (JGR), Modolo et al., 2017 (PSS), ... For magnetosheath asymmetries, see the works with Cluster data of Génot et al., and with ISEE data of Tatrallyay et al. For the discussions on Alfven Mach number effects see Lavraud & Borovsky, 2008.

- Foreshock effects: it seems to me that the foreshock effects are over emphasized. Actually the perturbations linked to turbulence processes in the magnetosheath are more directly connected to effects associated to the physics of the parallel shock than to the foreshock itself which lies upstream of the shock. In that respect I disagree with the last sentence of the abstract and similar statements in the paper (for instance l353). Could the authors demonstrate why the foreshock is so important and for which effects it should be distinguished with the parallel shock?

- Kinetic effects: on l300 simulation results on density asymmetry are opposed to those coming from an analysis of MHD equations. The authors point to kinetic effects. Why is it that kinetic effects matter specifically on this issue and not on other where simulations and MHD match? This requires more discussion. Even though this may be outside the scope of the paper, a comparison with 3D MHD simulation (for instance available at CCMC) would help pointing to specific kinetic effects inherent to the Vlasiator code.

- Global approach: the model is 2D in space and the magnetopause is not completely resolved such that a model magnetopause needs to be used. This puts limitation on the term "global" to qualify the simulations. I wonder if the compression/expansion in this limited 2D space can be adequately compared with the real 3D situation. Could the authors discuss this aspect? or point to literature as this has surely been already addressed.

- Scales: could the authors give information on the temporal and spatial scales resolved in the simulations? And compare them to typical scales like inertial lengths and

typical periods (inverses of plasma/cyclotron frequencies). How does this compare with the 150s used for averaging magnetosheath parameters? This would help the interpretation of density variability mentioned l289 for instance.

- Set-up: it is not clear to me why run 1 is set up in the XZ plane and arguments are sought for to justify it mimics correctly the XY plane. Why not using a proper set up in the XY plane from the start?

- Observations: for comparing observations and simulations the same statistical methodology should be employed, i.e. median or average for both, contrary to what is done in the paper.

Minor points:

- l95: 'warranted'. Do the authors mean 'mandatory'?

- Figure 1: mismatch between central / outer legends and d and e labels.

- l400: snaller

- l427: 'statistical'. Do the authors refer to observations here?

---

## Referee Comment (RC4) · Anonymous Referee #4 · 13 May 2020

The paper studies asymmetry in the Earth's dayside magnetosheath using global hybrid-Vlasov simulations and compares numerical results with a statistical dataset of THEMIS observations. The paper is clearly written and the results are new and interesting. However, some details about modeling are missed. I partly agree with the comments of three other reviewers and mention several important points from their reports below. I could recommend the paper for publication after major revision.

**Major remarks**

1. Although the Vlasiator model is well known and I believe it has been thoroughly described in the literature, the paper should provide more details on the runs under

discussion. In particular, (as also mentioned by one of the reviewers) the paper says nothing about spatial resolution. It would be useful to compare the resolution with the ion inertial length and gyroradius. The authors have already answered this issue in their reply to Reviewer 1 and I suppose it will appear in the paper too. The paper does not describe the simulation domains in each case; it only mentions that their size is different between the runs. I would be also curious to know what happens if the simulated intervals in Runs 2A and 2B would be increased since now they are shorter than in Run 1.

2. Both the Reviewers 1 and 3 noted that comparison with MHD runs for exactly the same solar wind conditions will be useful because this would emphasize which variations in the magnetosheath downstream of the quasi-parallel bow shock are essentially kinetic structures and cannot be predicted by MHD models. However, I do not think that it is necessary to run all MHD models available from CCMC, but it would be enough to make three runs with at least one model (e.g. SWMF/BATSRUS).

3. I also note that the solar wind conditions in the hybrid simulations are different from the typical solar wind conditions at L1. I am satisfied with the author's reply to Reviewer 1 that the Mach numbers in the solar wind stay in the typical interval and therefore the bow shock-magnetosheath properties may not be changed in comparison with those in observations. However, I would emphasize that the solar wind density of  $1 \text{ cm}^{-3}$  is significantly smaller than the average in observations (usually between 5 and  $10 \text{ cm}^{-3}$ ). I think the paper should clearly explain this because I guess that the low solar wind density may be a reason for the stronger fluctuations in the magnetosheath than those in the data.

4. Since the authors use average parameters both in the simulations and observations, I think it would be possible to add standard deviations to the figures, e.g. in the form of error bars. This would be helpful when comparing the differences between the runs (how significant is the difference with respect to the standard deviations). Besides, the authors mention in the text that they calculated longer time average intervals (line 290). How long are they and does this make any difference to their conclusions?

Minor remarks

1. The bibliography list in the paper is long, but I would like to mention two more papers, Zwan and Wolf (https://doi.org/10.1029/JA081i010p01636) who first mentioned the plasma depletion layer and Samsonov et al. (https://doi.org/10.1029/2000JA900150) who compared magnetosheath profiles downstream of the parallel and perpendicular bow shock using the anisotropic MHD model.

2. Line 83. "These processes would thus favour the quasi-parallel flank." But the results in the paper show the  $Q_{\perp}$ -favoured velocity asymmetry. How is this consistent? 3. Lines 149-152. The figures in the paper show that the spatial bins are asymmetric with respect to the Sun-Earth line. Please, explain how this asymmetry is taken into account if you use the same shape as Shue et al.'s model which is symmetrical.

4. Caption to Figure 2. Please, define  $\theta_{Bn}$ .

5. Lines 233-235. Is  $\theta_{Bn}$  equal to  $0^{\circ}$  and  $90^{\circ}$  near the terminator plane?

6. Lines 265-266. I think it is better "density compression ratio" instead of "shock compression ratio".

7. Label on Figure 5 says that the lines correspond to runs 1 & 2A and 2B but this contradicts the text (lines 265-266).

8. Figure 4. The author may add an arrow to indicate the stagnation point.

9. Lines 367-369. Is it better to say about an increase in the magnetic field on the quasi-perpendicular flank than about a decrease on the quasi-parallel flank?

---

## Author Comment (AC2) · 28 May 2020

**We thank the referee for their positive feedback on our manuscript and for their insightful remarks. Please find below our point-by-point response in bold font.**

This manuscript describes 2D simulations of the dayside magnetosheath using the hybrid Vlasiator code, for three different upstream conditions (one in the noon-midnight plane, and two in the GSE equatorial plane). The authors appropriately describe the capabilities as well as the issues and shortcomings pertaining to this hybrid model. The detailed description of the challenges of magnetosheath studies using spacecraft observations is also highly appropriate. The explanations provided regarding the nu-

[Figure]

merical results of magnetosheath asymmetries of parameters (B, density, and velocity) downstream of the Qpara and Qperp bow shock regions as a function of angle from the Sun-Earth line are plausible, though perhaps not the only possible explanations. Comparing the numerical simulation results with magnetosheath observations by the THEMIS spacecraft is also highly appropriate.

There are two significant concerns with the manner in which the study results are presented. These ought to be fairly easily addressed, but are important because they directly affect most of the figures and results presented in this study:

1) Magnetosheath parameters determined from the numerical simulations within each spatial bin and for the time interval used are presented as averages; whereas the magnetosheath parameters determined from spacecraft observations are presented as medians. In order to ensure that the comparisons between simulations and observations are meaningful, the same statistical measure should be used for both (ideally medians, to avoid outlier kinetic effects due to processes at the bow shock convected into specific magnetosheath bins from unduly influencing the overall average value). An alternative is to demonstrate that within the magnetosheath bins, the distribution of values used to determine the spatial and temporal average is Gaussian, so that the average and median values are the same.

**Thank you for pointing this out, we should indeed have used the same statistical measure to quantify the "global" value of the asymmetry in each run.**

**First, we would like to clarify that the magnetosheath parameters were computed using the same methodology both in the numerical simulations and the spacecraft observations, as averages inside each spatial bin. We chose to use averages so that our results are comparable to the statistical results presented in Dimmock et al. [2017]. We note however that Walsh et al. [2012] used median values inside the spatial bins rather than mean values. In order to check that our results are not sensitive to using median or mean values, we calculated the**

asymmetries based on the median value in each bin. We found that both the mean and the median yield very similar asymmetry levels.

Once we had obtained the asymmetry values for each azimuthal and radial bins, we looked for a means to quantify the "global" value of the asymmetry in each run. After carefully trying out both median and mean values as indicators of the "global" asymmetry level, we came to the conclusion that the large variation of the asymmetry level from bin to bin in each simulation makes both of these problematic. To give a better description of our results, in the revised manuscript, we will give instead the range of values for each asymmetry. For example the magnetic field asymmetry in the central magnetosheath in Run 1 ranges between 0 and 15%, and compare it with the 5-10% values in Dimmock et al. [2017].

2) It is difficult to judge the robustness of the results, because there are no estimates of the statistical spread (uncertainties) associated with the averages (or medians). From the simulations, sampling in appropriately sized sub-spatial and sub-temporal bins to provide e.g., standard deviations (or quartiles) used in the estimate of the asymmetry would instill considerable confidence that the percentage of asymmetry results are robust. Similarly for the THEMIS observations, it would be more appropriate if statistical estimates representing the range of values within each bin are determined and then used to estimate the range of values (measure of uncertainty) for the percentages of asymmetry for the various plasma parameters.

Thank you for this suggestion. In the revised manuscript, we will add error bars to the asymmetry plots (line plots in Figures 1, 3, 6 and 7). As done in Dimmock et al. [2017], we estimate the error on the magnetosheath parameters as the standard error of the mean (standard deviation divided by the square root of the size of the bin sample). We then use this error to calculate the minimum and maximum values of the asymmetry in each bin, which determines the extent of the error bars in the asymmetry plots.
**The two figures below show two examples of the updated figures which will be included in the revised manuscript. The error bars in our numerical results are much smaller than for the observational spacecraft data set, most likely because of the steady upstream conditions in our simulations.**

Minor issues:

Line 268: considerable -> considerably

Figure 4: Should label which side of the plot corresponds to Qpara, and which side corresponds to Qperp.

Line 341: magnetosheah -> magnetosheath

**Thank you for picking up these typos, we will correct them in the revised manuscript. We will add the labels on Figure 4.**
* * *
[Figure]

**Magnetic field strength (central magnetosheath)**

$Q_\perp$-favoured asymmetry

$Q_\parallel$-favoured asymmetry

Run 1
Run 2A
Run 2B

Asymmetry [%]

Angle from Sun-Earth line [°]

**Fig. 1.** Magnetic field asymmetry in the central magnetosheath (simulations) (Fig 1e – should be 1d in the manuscript)

**Magnetic field strength (central magnetosheath)**

Q$_\perp$-favoured asymmetry

Legend:
- $40° < \theta_{Bx} < 50°$
- $20° < \theta_{Bx} < 35°$

Q$_\parallel$-favoured asymmetry

Y-axis: **Asymmetry [%]**

X-axis: **Angle from Sun-Earth line** [°]

**Fig. 2.** Magnetic field asymmetry in the central magnetosheath (observations) (Fig. 7a in the manuscript)

---

## Author Comment (AC5) · 29 May 2020

Contrary to what we indicated in our response to the reviewer's comment, the spatial resolution in Run 1 is 300 km, that is, 1.3 ion skin depth, and not 228 km. Runs 2A and 2B have a spatial resolution of 228 km. We apologize for the confusion.

---

## Author Response (AR1)

**Response to Reviewer #1**

We thank the reviewer for reviewing our manuscript and for providing detailed comments. The reviewer raised in particular several major concerns regarding the methodology employed in our study, which we address in detail below. As can be seen from our response, we feel that these issues can be resolved by better clarifying our methodology in the revised version of our manuscript.

Our responses are marked in **bold** font in the text below, in between the reviewer's comments.

Summary of the manuscript:

Authors have studied magnetosheath dawn-dusk asymmetries of the magnetic field (B), proton number density (n) and plasma flow speed(V) in the 2-D global Vlasiator simulation and compared the results with statistics from the THEMIS observations. The main conclusion made by the authors is that while the polarity of the asymmetries agrees with the THEMIS results, the magnitudes are in disagreement because Vlasiator is run with one set of conditions, whereas spacecraft observations show cumulation of the observations from different solar wind conditions and IMF orientations.

**Overall evaluation of the manuscript:**

Unfortunately, in its present form I am unable to recommend this manuscript for publication in the scientific literature for the following reasons:

1) There are no original ideas or new scientifically valid results. There have been several past studies of the magnetosheath properties using both spacecraft data and different plasma approximation (e.g., MHD, hybrid, kinetic-simulations). Some of these have been cited but not all: The bow shock is not the only source for magnetosheath plasma. Also magnetopause processes are important that can transport magnetospheric particles into the magnetosheath. The study of the dawn-dusk asymmetries of B, n and V using spacecraft data has already been done. If the motivation is to solely test the code robustness, by using the study of dawn-dusk asymmetries as a validation effort against spacecraft data, a technical paper may be more suitable.

We agree that there has been a number of observational studies of the dawn-dusk asymmetries of the parameters we selected for our study, as well as a few numerical studies using MHD models. However, we politely disagree with the reviewer regarding the lack of new results in our study.

In this manuscript, we present the first study of these asymmetries using a global hybrid-Vlasov model, which provides us with new information regarding those asymmetries. To our knowledge, this is the first in-depth quantification of these asymmetries using a hybrid-kinetic model. Part of our manuscript is indeed dedicated to validating our simulation results based on the comparison with previous works, but we also present novel results:

(1) We show that foreshock kinetic processes have a strong impact on the magnetosheath density, thus providing an answer to the long-standing question of the variability of this asymmetry, and reconciling the vastly different results obtained in previous studies [Paularena et al., 2001; Walsh et al., 2012; Dimmock et al., 2016]

(2) We investigate the influence of the IMF cone angle on the asymmetries, which has not been studied before, neither with spacecraft measurements nor with models, and show that the magnetic field asymmetry and the variability of the magnetosheath density increase when the cone angle is reduced from 45° to 30°.

(3) We investigate the influence of the Alfvén Mach number on the asymmetries, in a range of Mach numbers that is not easily accessible with observations, and show that the variability of the magnetosheath density and velocity is reduced at low Alfvén Mach numbers.

In the revised manuscript, we have reformulated part of the abstract (lines 12-13 and 17-21) and of the conclusions (lines 508-510) to better highlight these novel results.

Regarding the importance of magnetopause processes, we agree with the reviewer that they can indeed affect magnetosheath properties near the magnetopause. In the revised manuscript, we added a brief mention to magnetopause processes (lines 40-43).

2) For scientific paper "more in depth" analysis of the physical mechanisms is required to address what differences are due to numerical issues, what are due to kinetic physics, and what are the mechanisms. For example, how are the magnetosheath densities > 4 explained, if solar wind density is 1? How are these results affected by grid-resolution.

According to MHD theory, the compression at the bow shock should indeed result in the magnetosheath density just downstream of the shock being at most four times larger than the solar wind density. This is what is observed downstream of the quasi-perpendicular portion of the bow shock in our numerical simulations (see the upper half of the top three panels in Fig. 6), where MHD theory mostly holds. At the quasi-parallel shock, on the other hand, kinetic processes become most prominent, and larger densities can be observed in the magnetosheath.

These large densities come essentially from density fluctuations in the foreshock, resulting in upstream densities that are already well above the plasma density in the pristine solar wind [see, for example, the numerical simulations presented in Omidi et al., 2014 and Turc et al., 2018; and the spacecraft measurements presented in the review by Eastwood et al., 2005]. When these patches of high density cross the bow shock, their fourfold compression results in downstream densities that exceed four times the solar wind density.

Such large densities in the magnetosheath, above the MHD limit, are a common feature in hybrid-kinetic simulations of the bow shock/magnetosheath system [see for example Figs. 9 and 10 in Omidi et al., 2014; Fig. 7 in Karimabadi et al., 2014]. We have added a few sentences in Section 3.3 of the manuscript (lines 343-346) to compare the magnetosheath density in our simulations with that in these previous numerical works.

Regarding the possible effects of grid resolution, the scale of these high density patches in the magnetosheath is much larger than the cell size, and thus they shouldn't be affected by the finite resolution in our simulations. Grid resolution, on the other hand, can impact which wave modes develop in the simulation [see Dubart et al., 2020, pre-print available in Annales Geophysicae Discussions: https://www.ann-geophys-discuss.net/angeo-2020-24/]. Previous works have shown that the compressional foreshock waves, the so-called 30 s waves, which are responsible for large-scale density fluctuations in the foreshock, are properly resolved in Vlasiator and their properties match those obtained from spacecraft measurements [Palmroth et al., 2015; Turc et al., 2018, 2019]. Also, foreshock transients such as cavitons and spontaneous hot flow anomalies, which can also result in density variations, develop as expected in the simulation [Blanco-Cano et al., 2018]. Therefore, the spatial resolution of our simulation is not expected to affect the plasma compression at the bow shock.

More discussion regarding the resolution is given below, as a response to another of the referee's comments. We have also added some discussion regarding the grid resolution in the revised manuscript (lines 444-454).

3) The Methodology of the present paper is flawed so no accurate scientific conclusions can be made at this time. The authors' conclusion, that the disagreement with the data and simulation is due to the fact that simulation is run for one set of conditions, whereas spacecraft data contains the history of IMF and solar wind, is only one possible reason. Paper fails to discuss the other, more plausible and likely more significant reasons listed below:

i) Code is run with 750 km/s solar wind speed. This condition occurs rarely as the average solar wind speed is slightly less than 400 km/s. Therefore, it is not logical or scientifically justified to compare the runs with 750 km/s solar wind velocity with the spacecraft statistics collected in the magnetosheath, when the solar wind flow preceding the THEMIS data collection is about 400 km/s. With higher solar wind speeds, the shock compression becomes stronger and one would expect higher magnetic field strengths downstream of the quasi-perpendicular shock than for solar wind speeds of 400 km/s.

While we agree with the reviewer that such fast speeds are rarely encountered in the solar wind at Earth, we will demonstrate that running the model with a 750 km/s solar wind speed is not an issue for the present study, and that our methodology is valid.

Firstly, the key parameter controlling the shock compression is the shock Mach number [e.g., Treumann et al., 2009], which indeed depends on the solar wind velocity, but also on other parameters such as the density, temperature and magnetic field strength, in the case of the magnetosonic Mach number. In our simulation, the upstream parameters are chosen such that the resulting Alfvén and magnetosonic Mach numbers have typical values for the solar wind at Earth: in our Runs 1 and 2A, the Alfvén Mach number  $M_A$  is 6.9 and the magnetosonic Mach number  $M_{ms}$  is 5.5, and our low Mach number run has  $M_A = 3.4$  and  $M_{ms} = 3.3$ . Typical values for these Mach numbers at Earth are  $2.5 < M_A < 12$  and  $2 < M_{ms} < 7$  [Winterhalter & Kivelson, 1988]. Therefore, despite the large solar wind speed, we have a typical compression ratio at the bow shock with our input parameters (see Fig. 5). We will add a sentence in Section 2.2, where the simulation runs are described, to indicate that the Mach number values in our runs are typical values at Earth.

Second, we would like to point out that all observational studies of magnetosheath asymmetries rely on the assumption that the variation of the shock compression is relatively small in the range of solar wind conditions encountered at Earth, and thus that magnetosheath parameters can be normalised to their solar wind counterparts to obtain the average distribution of magnetosheath properties. This normalisation is essential to observational studies, which are based on compilations of spacecraft measurements in the magnetosheath associated with a wide variety of solar wind conditions. Global maps of normalised magnetosheath parameters are presented for example in Paularena et al. [2001], Longmore et al. [2005] and Dimmock et al. [2013-2017]. Walsh et al. [2012] present both raw and normalised magnetosheath parameters, and the latter are more narrowly-distributed around the results from MHD simulations. The only instance in which Dimmock et al. [2017] do not use a normalised parameter is for the ion temperature, because of cross-calibration issues when comparing temperature measurements from different spacecraft (THEMIS in the magnetosheath and ACE or Wind in the solar wind). Moreover, Dimmock et al. [2017] compare the levels of magnetosheath asymmetries for solar wind velocities below and above 400 km/s (splitting their data set into two halves), and find no significant change in the asymmetries between these two ranges of solar wind velocities.

Large solar wind speeds may strongly affect the flank magnetosheath parameters, as large solar wind speeds are conducive to the development of the Kelvin-Helmholtz instability (KHI) at the magnetopause [e.g. Kavosi & Raeder, 2015]. However, our study concentrates on magnetosheath asymmetries in the plane containing the IMF vector, while KHI develops in the plane that is perpendicular to the IMF (e.g. in the equatorial plane for a northward IMF). Therefore, our results would not be affected by the KHI.

Therefore, the large solar wind speeds in our simulations are not an issue to quantify the magnetosheath asymmetry levels away from the magnetopause, and the normalisation of the data to the solar wind quantities together with the typical shock Mach numbers and compression ratio in our simulations ensure that the comparison with spacecraft observations is relevant. In Section 2.3 of the revised manuscript, we now explain in more detail our approach and its validity. ii) The runs use upstream solar wind density of 1/cc, which would result in a maximum downstream density of about 4/cc downstream of quasi-perp shock. Assuming the values of 1- 4/cc, the ion inertial length would be 228 km to 114 km in the magnetosheath, respectively, and for higher densities these scales get even smaller. The paper does not describe what the coordinate-space resolution is used for the runs. In order to appropriately resolve the physical ion scales, the Vlasiator should use about 5 to 10 cells/ion inertial length, which requires coordinate space resolution at the minimum of about 20-40 km, otherwise the kinetic effects can artificially dominate at larger length scales than in the real system. Furthermore, if the ion inertial scale at the bow shock is not appropriately resolved, the results pertaining to kinetic shock physics are physically meaningless or exaggerated.

The resolution of our grid in ordinary space is 227 km, which corresponds to one ion inertial length in the solar wind. As shown by numerous works, this resolution is sufficient to capture most ion kinetic processes in a simulation, and yields results that are in good agreement with spacecraft observations. In this, we respectfully disagree with the reviewer's statement that one would require 5-10 cells per ion inertial length in hybrid kinetic simulations.

We kindly refer the reviewer to the works of Omidi and colleagues [e.g., Omidi et al., 2014, 2016] and Blanco-Cano and colleagues [e.g., Blanco-Cano et al., 2006, 2009], in which a cell size of 1 solar wind ion inertial length is also used, allowing detailed investigations of ion kinetic processes in the foreshock and the magnetosheath. In the simulations of Shi et al. [2013, 2017] the spatial resolution is 1 to 2 cells per ion inertial length, depending on the position in the simulation domain. Karimabadi et al. [2014] present the results of 7 runs, most of them with a resolution of 2 cells per ion inertial length. The run with the best resolution has 4 cells per ion inertial length, but at the expense of the size of the simulation domain.

The results of these models (Vlasiator included) pertaining to ion kinetic processes have been extensively validated against spacecraft measurements [e.g., Sibeck et al., 2008, Blanco-Cano et al., 2011, Palmroth et al., 2015, Turc et al., 2019]. They have predicted kinetic phenomena such as foreshock bubbles [Omidi et al., 2010] and cavitons [Blanco-Cano et al., 2009] that have been later on been confirmed in spacecraft measurements [Archer et al., 2015; Kajdic et al., 2011].

At the shock front, a cell size of 1 ion inertial length or larger may not correctly evaluate the gradient in the ramp. However, the hybrid-Vlasov formalism based on distribution functions enables the use of a slope limiter which allows for total variation diminishing evolution of discontinuities and steep slopes even at somewhat lower resolution. The shock transition is therefore well described in our simulations, and the downstream parameters are correctly modelled.

The study we present here focuses on the large-scale distribution of magnetosheath properties. Therefore, the spatial resolution of 1 cell per ion inertial length in our Vlasiator runs is sufficient to study global magnetosheath parameters and how they are impacted by ion kinetic physics. We have added a discussion on the spatial resolution in our simulations in the revised version of the manuscript (lines 444-454).

iii) Since the B, n and V are MHD quantities, authors should also run their case exactly with same parameters using the major global, state-of-the-art 3-D MHD codes available through Community Coordinated Modeling Center and compare their Vlasiator results with these.

Magnetosheath asymmetries have been already studied using MHD models in previous studies, as described in the Introduction [Walsh et al., 2012; Dimmock et al., 2013]. The results of Walsh et al. [2012] show in particular that the magnitude of the asymmetries is larger in their MHD simulations than in the observations. Their interpretation is similar as

ours: the asymmetries are larger in the simulations because they are run for a single IMF orientation, while the observations combine many different IMF orientations.

We have expanded the comparison to previous MHD simulations in the Discussion section of the revised manuscript (lines 467-469), but we do not feel that running new MHD simulations will bring novel information that is not already reported in the literature.

**Recommendation:**

While I cannot recommend this paper for publication at this time, I hope the authors will use this feedback as an opportunity to improve the paper, and for transparency include additional missing information (e.g., the grid-resolution etc.). This work requires significant further code validation efforts in a global scale, where the effects of spatial grid-resolution are systematically studied, so that the results can be correctly interpreted.

Please see the specific comments below that need to be addressed after which the paper may eventually become suitable for scientific literature:

We thank the reviewer for this very detailed set of suggestions on how to further proceed with this study. However, we believe that our methodology is appropriate for the present study, and that our conclusions are well supported by the analysis of our numerical results, as detailed in our responses to the reviewer's previous comments (see above).

Incidentally, we would like to mention that a typical Vlasiator run requires from a few to tens of million CPU hours. Performing a new set of runs as suggested by the reviewer is extremely costly, and thus cannot be done on a short notice, as it requires months of planning, application for computing resources, running and validation of the model's outputs.

Specific comments:

1. Calculate the statistical solar wind and IMF condition from THEMIS statistics used in the paper.

2. Plot and show a spatial map in x-y -plane for each run of the i) ion inertial length, ii) ion gyroradius and iii) plasma beta, and collect a mean, minimum and maximum values of these at the central magnetosheath where the statistics pertaining to study is being collected during the course of the simulation.

3. Re-run Vlasiator with the statistical conditions and with the appropriate resolution (whichever length-scale is the smallest). For plasma beta of 1, the both length-scales should be the same. Compare the results with those in the present manuscript.

4. Add details and benchmarking how the phase space velocity distributions are processed to calculate n and V in 2-D-plane (as one can cut a 3-D velocity distribution in infinite ways). Show how the processing of the velocity distribution functions affects the results. Convince the reader of the validity of this processing at different regions in the magnetosheath.

While our simulation domain is 2D in ordinary space, the velocity space, in which the velocity distribution functions evolve, is 3D in Vlasiator. Therefore, no 2D cut is performed in the ion velocity distribution functions. The density and velocity are calculated as the zeroth and first velocity moments of the distribution functions, i.e., by integrating the distribution function over the velocity space, based on plasma kinetic theory. This approach is valid everywhere in the simulation domain. To clarify this, we have added a sentence in Section 2.1 explaining that the macroscopic plasma parameters are obtained by integrating the velocity distribution functions (lines 136-138).

5. Re-run Vlasiator with a) old-set of parameters shown in this manuscript while appropriately resolving these length-scales and compare with the results from original resolution.

6. Ion and electron temperatures are an important quantity to demonstrate the dawn-dusk asymmetry. It will be very interesting to see these two parameters, and the perpendicular and parallel temperatures.

Vlasiator is a hybrid model, which describes ion kinetic physics but treats electrons as a fluid. Therefore, it cannot be used to study electron temperatures in the magnetosheath. As concerns the ion temperature, we decided to leave it for future work, as we think its investigation warrants a paper of its own (as was done for example by Dimmock et al., 2015 for the ion temperature asymmetry in the magnetosheath as observed by the THEMIS spacecraft). Here, we chose to concentrate on the magnetic field strength, plasma density and bulk velocity, and how they vary as a function of the cone angle and the Alfvén Mach number.

7. Since the majority of the quantities compared in this study can also be obtained by the MHD global simulation, I suggest the authors also run the same solar input in the CCMC to compare with all the other three major MHD models and show the advantage of Vlasiator results.

**As mentioned earlier, MHD simulations of magnetosheath asymmetries have already been performed by Walsh et al. [2012] and provide results consistent with our findings.**

8. What is the "zebra stripes" structure on the dusk side, are those physical waves (which wave-mode) or a grid oscillation?

These are physical oscillations, which are well resolved in the simulation as their wavelength is significantly larger than the cell size. The properties of these oscillations are consistent with that of the overshoot of the quasi-perpendicular bow shock, which has been observed for example by the ISEE and the Cluster spacecraft [Livesey et al., 1982; Bale et al., 2005]:

- their amplitude decays when moving further away from the shock front [Saxena et al., 2005]. - their wavelength is related to the ion gyroradius [Saxena et al., 2005]. This is evidenced by their smaller scale in Run 2B, where the IMF strength is doubled, and thus the gyroradius is smaller.

- their amplitude is related to the upstream Mach number [Livesey et al., 1982]. It is much smaller in Run 2B (low  $M_A$  run), where the oscillations are barely visible in Figs 1, 3 and 6, than in the other two runs.

Investigating in more detail the properties of these oscillations and their relevance for magnetosheath transport processes is a study of its own, which is currently under way.

9. From MHD the maximum shock compression ratio would give magnetosheath densities of 4/cc if the density in the solar wind is 1/cc. Here the maximum density in the magnetosheath is 6/cc. Is this a kinetic effect and what is the physical mechanism to generate that? How is the area of the > 4/cc density regions in the magnetosheath dependent on the ion gyro-radius/inertial scale and grid resolution when compared to MHD simulations that are run with the same parameters and same resolution?

As discussed above, these high density patches in the magnetosheath are most likely due to density enhancements in the foreshock which are further compressed upon crossing the bow shock. The density enhancements in the foreshock are due to compressional waves and transient structures. These phenomena are not described in MHD models, as the foreshock is inherently a kinetic structure. These results are therefore not comparable with MHD, where such phenomena do not exist.

The scale of these density enhancements is related to that of the foreshock ULF waves waves, which modulate the foreshock parameters, including the density [e.g., Blanco-Cano et al., 2006; Turc et al., 2018]. In previous studies, we have shown that the properties of these waves in our simulations are described at their correct scales and are in excellent agreement with spacecraft observations [Palmroth et al., 2015; Turc et al., 2018, 2019].

Minor comments:

1. The Discussion and Conclusions are repetitive. This could be made more concise.

**We feel that the content of these two sections is appropriate and that they are complementary rather than repeating each other. As no concrete suggestions for shortening were given, we were not able to identify which parts the reviewer found redundant.**

2. The referencing is inadequate. Authors should extend their citations to include some of the following. Please see previous global hybrid simulation studies of the magnetosheath (e.g. by Y. Lin et al. (2001-2020), H. Karimabadi et al., N. Omidi et al,) and several missing studies related to spacecraft observations and statistics of the various foreshock transient that modify magnetosheath properties (e.g., F. Plaschke et al. (2013-2019), D. Turner at al., H. Hietala et al. (2009-2018), T. Liu et al.(2017-2019), H. Zhang et al. (2013-,) as well as leakage of magnetosheatic particles into the magnetosheath by various processes (e.g., I. Cohen et al. 2017; K. Sorathia et al., 2019), and due to local magentosheath physics (e.g., P. Gary et al., 2006 A. Retino et al. 2007, D. Sundkvist et al. 2007, J. Soucek et al.(2008-2015), V. Genot et al.(2001-2009), T. Phan et al., 2018).

We have added some of these references regarding magnetosheath properties in our introduction. We have also added a new paragraph in our introduction describing previous hybrid simulations, which include some of the references listed above (lines 111-118).

**References (not previously included in the manuscript bibliography)**

Archer, M. O., Turner, D. L., Eastwood, J. P., Schwartz, S. J., & Horbury, T. S.: Global impacts of a Foreshock Bubble: Magnetosheath, magnetopause and ground-based observations, Planetary and Space Science, 106, 56, doi:10.1016/j.pss.2014.11.026, 2015.

Bale, S. D., Balikhin, M. A., Horbury, T. S., Krasnoselskikh, V. V., Kucharek, H., Möbius, E., Walker, S. N., Balogh, A., Burgess, D., Lembège, B., Lucek, E. A., Scholer, M., Schwartz7 10, S. J., & Thomsen, M. F.: Quasi-perpendicular Shock Structure and Processes, Space Science Reviews, 118, 161, doi:10.1007/s11214-005-3827-0, 2005.

Blanco-Cano, X., Omidi, N., & Russell, C. T.: Macrostructure of collisionless bow shocks: 2. ULF waves in the foreshock and magnetosheath, Journal of Geophysical Research (Space Physics), 111, A10205, doi:10.1029/2005JA011421, 2006.

Blanco-Cano, X., Omidi, N., & Russell, C. T.: Global hybrid simulations: Foreshock waves and cavitons under radial interplanetary magnetic field geometry, Journal of Geophysical Research (Space Physics), 114, A01216, doi:10.1029/2008JA013406, 2009.

Blanco-Cano, X., Kajdič, P., Omidi, N., & Russell, C. T.: Foreshock cavitons for different interplanetary magnetic field geometries: Simulations and observations, Journal of Geophysical Research (Space Physics), 116, A09101, doi:10.1029/2010JA016413, 2011.

Blanco-Cano, X., Battarbee, M., Turc, L., Dimmock, A. P., Kilpua, E. K. J., Hoilijoki, S., Ganse, U., Sibeck, D. G., Cassak, P. A., Fear, R. C., Jarvinen, R., Juusola, L., Pfau-Kempf, Y.,

Vainio, R., & Palmroth, M.: Cavitons and spontaneous hot flow anomalies in a hybrid-Vlasov global magnetospheric simulation, Annales Geophysicae, 36, 1081, doi:10.5194/angeo-36-1081-2018, 2018.

Dubart, M., Ganse, U., Osmane, A., Johlander, A., Battarbee, M., Grandin, M., Pfau-Kempf, Y., Turc, L., and Palmroth, M.: Resolution dependence of magnetosheath waves in global hybrid-Vlasov simulations, Ann. Geophys. Discuss., https://doi.org/10.5194/angeo-2020-24, in review, 2020.

Eastwood, J. P., et al.: The Foreshock, Space Science Reviews, 118, 41, doi:10.1007/s11214-005-3824-3, 2005.

Kajdič, P., Blanco-Cano, X., Omidi, N., & Russell, C. T.: Multi-spacecraft study of foreshock cavitons upstream of the quasi-parallel bow shock, Planetary and Space Science, 59, 705, doi:10.1016/j.pss.2011.02.005, 2011.

Karimabadi, H., Roytershteyn, V., Vu, H. X., Omelchenko, Y. A., Scudder, J., Daughton, W., Dimmock, A., Nykyri, K., Wan, M., Sibeck, D., Tatineni, M., Majumdar, A., Loring, B., & Geveci, B.: The link between shocks, turbulence, and magnetic reconnection in collisionless plasmas, Physics of Plasmas, 21, 062308, doi:10.1063/1.4882875, 2014.

Kavosi, S., & Raeder, J.: Ubiquity of Kelvin-Helmholtz waves at Earth's magnetopause, Nature Communications, 6, 7019, doi:10.1038/ncomms8019, 2015.

Livesey, W. A., Kennel, C. F., & Russell, C. T.: ISEE-1 and -2 observations of magnetic field strength overshoots in quasi-perpendicular bow shocks, Geophysical Research Letters, 9, 1037, doi:10.1029/GL009i009p01037, 1982.

Omidi, N., Eastwood, J. P., & Sibeck, D. G.: Foreshock bubbles and their global magnetospheric impacts, Journal of Geophysical Research (Space Physics), 115, A06204, doi:10.1029/2009JA014828, 2010.

Omidi, N., Sibeck, D., Gutynska, O., & Trattner, K. J.: Magnetosheath filamentary structures formed by ion acceleration at the quasi-parallel bow shock, Journal of Geophysical Research (Space Physics), 119, 2593, doi:10.1002/2013JA019587, 2014.

Omidi, N., Berchem, J., Sibeck, D., & Zhang, H.: Impacts of spontaneous hot flow anomalies on the magnetosheath and magnetopause, Journal of Geophysical Research (Space Physics), 121, 3155, doi:10.1002/2015JA022170, 2016.

Palmroth, M., et al.: ULF foreshock under radial IMF: THEMIS observations and global kinetic simulation Vlasiator results compared, Journal of Geophysical Research (Space Physics), 120, 8782, doi:10.1002/2015JA021526, 2015.

Saxena, R., Bale, S. D., & Horbury, T. S.: Wavelength and decay length of density overshoot structure in supercritical, collisionless bow shocks, Physics of Plasmas, 12, 052904, doi:10.1063/1.1900093, 2005.

Shi, F., Lin, Y., & Wang, X.: Global hybrid simulation of mode conversion at the dayside magnetopause, Journal of Geophysical Research (Space Physics), 118, 6176, doi:10.1002/jgra.50587, 2013.

Shi, F., Cheng, L., Lin, Y., & Wang, X.: Foreshock wave interaction with the magnetopause: Signatures of mode conversion, Journal of Geophysical Research (Space Physics), 122, 7057, doi:10.1002/2016JA023114, 2017.

Sibeck, D. G., Omidi, N., Dandouras, I., & Lucek, E.: On the edge of the foreshock: modeldata comparisons, Annales Geophysicae, 26, 1539, doi:10.5194/angeo-26-1539-2008, 2008.

Turc, L., Roberts, O. W., Archer, M. O., Palmroth, M., Battarbee, M., Brito, T., Ganse, U., Grandin, M., Pfau-Kempf, Y., Escoubet, C. P., & Dandouras, I.: First Observations of the Disruption of the Earth's Foreshock Wave Field During Magnetic Clouds, Geophysical Research Letters, 46, 12,644, doi:10.1029/2019GL084437, 2019.

Winterhalter, D., & Kivelson, M. G.: Observations of the Earth's bow shock under high Mach number/high plasma beta solar wind conditions, Geophysical Research Letters, 15, 1161, doi:10.1029/GL015i010p01161, 1988.

**Response to Reviewer #2**

**We thank the referee for their positive feedback on our manuscript and for their insightful remarks. Please find below our point-by-point response in **bold** font.**

This manuscript describes 2D simulations of the dayside magnetosheath using the hybrid Vlasiator code, for three different upstream conditions (one in the noon-midnight plane, and two in the GSE equatorial plane). The authors appropriately describe the capabilities as well as the issues and shortcomings pertaining to this hybrid model. The detailed description of the challenges of magnetosheath studies using spacecraft observations is also highly appropriate. The explanations provided regarding the numerical results of magnetosheath asymmetries of parameters (B, density, and velocity) downstream of the Qpara and Qperp bow shock regions as a function of angle from the Sun-Earth line are plausible, though perhaps not the only possible explanations. Comparing the numerical simulation results with magnetosheath observations by the THEMIS spacecraft is also highly appropriate.

There are two significant concerns with the manner in which the study results are presented. These ought to be fairly easily addressed, but are important because they directly affect most of the figures and results presented in this study:

1) Magnetosheath parameters determined from the numerical simulations within each spatial bin and for the time interval used are presented as averages; whereas the magnetosheath parameters determined from spacecraft observations are presented as medians. In order to ensure that the comparisons between simulations and observations are meaningful, the same statistical measure should be used for both (ideally medians, to avoid outlier kinetic effects due to processes at the bow shock convected into specific magnetosheath bins from unduly influencing the overall average value). An alternative is to demonstrate that within the magnetosheath bins, the distribution of values used to determine the spatial and temporal average is Gaussian, so that the average and median values are the same.

Thank you for pointing this out, we should indeed have used the same statistical measure to quantify the "global" value of the asymmetry in each run.

We would like to clarify that the magnetosheath parameters were computed using the same methodology both in the numerical simulations and the spacecraft observations, as averages inside each spatial bin. We chose to use averages so that our results are comparable to the statistical results presented in Dimmock et al. [2017]. We note however that Walsh et al. [2012] used median values inside the spatial bins rather than mean values.

In order to check that our results are not sensitive to using median or mean values, we calculated the asymmetries based on the median value in each bin. We found that both the mean and the median yield very similar asymmetry levels. This is now mentioned in the revised manuscript at lines 206-210.

After carefully trying out both median and mean values as indicators of the "global" asymmetry level, we came to the conclusion that the large variation of the asymmetry level from bin to bin in each simulation makes both of these problematic. To give a better description of our results, in the revised manuscript, we have given instead the range ofvalues for each asymmetry. For example the magnetic field asymmetry in the central magnetosheath in Run 1 ranges between 0 and 15%, and compare it with the 5-10% values in Dimmock et al. [2017] (see lines 266-269).

2) It is difficult to judge the robustness of the results, because there are no estimates of the statistical spread (uncertainties) associated with the averages (or medians). From the simulations, sampling in appropriately sized sub-spatial and sub-temporal bins to provide e.g., standard deviations (or quartiles) used in the estimate of the asymmetry would instill considerable confidence that the percentage of asymmetry results are robust. Similarly for the THEMIS observations, it would be more appropriate if statistical estimates representing the range of values within each bin are determined and then used to estimate the range of values (measure of uncertainty) for the percentages of asymmetry for the various plasma parameters.

Thank you for this suggestion. In the revised manuscript, we have added error bars to the asymmetry plots (line plots in Figures 1, 3, 6 and 7). As done in Dimmock et al. [2017], we estimate the error on the magnetosheath parameters as the standard error of the mean (standard deviation divided by the square root of the size of the bin sample). We then use this error to calculate the minimum and maximum values of the asymmetry in each bin, which determines the extent of the error bars in the asymmetry plots. The estimation of the error is described at lines 210-212 and 226-228.

The error bars in our numerical results are much smaller than for the observational spacecraft data set, most likely because of the steady upstream conditions in our simulations.

Minor issues:

Line 268: considerable -> considerably Figure 4: Should label which side of the plot corresponds to Qpara, and which side corresponds to Qperp. Line 341: magnetosheah -> magnetosheath

Thank you for picking up these typos, we have corrected them in the revised manuscript. We have added the suggested labels on Figure 4.

**Response to Reviewer #3**

**We thank the reviewer for their careful examination of our manuscript and their constructive comments. Please find below our point-by-point response in **bold** font.**

The paper describes the Earth magnetosheath response to the solar wind inflow using the Vlasiator code. The focus is put on the various asymmetries of plasma and magnetic parameters in three cases with varying IMF orientation and Alfven Mach number. The results are then compared to an analysis of THEMIS observations which was published previously (Dimmock et al.'s papers). The objectives are sound, the code and the analysis appropriate, however a number of key points make the paper not mature enough in the present form. They are listed first, then minor issues follow.

**Major points:**

- References: the references to previous works are not adequate. Concerning hybrid codes for the magnetosheath, the literature was already vast before Vlasiator and 6D simulations of solar wind / planetary plasma interactions exist, e.g. Travnicek et al., 2007 (GRL), Hercik et al., 2013 (JGR), Modolo et al., 2017 (PSS), ... For magnetosheath asymmetries, see the works with Cluster data of Génot et al., and with ISEE data of Tatrallyay et al. For the discussions on Alfven Mach number effects see Lavraud & Borovsky, 2008.

**Thank you for providing these references, we have added them to the introduction in the revised manuscript (lines 68-70 and 111-118).**

- Foreshock effects: it seems to me that the foreshock effects are over emphasized. Actually the perturbations linked to turbulence processes in the magnetosheath are more directly connected to effects associated to the physics of the parallel shock than to the foreshock itself which lies upstream of the shock. In that respect I disagree with the last sentence of the abstract and similar statements in the paper (for instance 1353). Could the authors demonstrate why the foreshock is so important and for which effects it should be distinguished with the parallel shock?

We emphasized the importance of the foreshock because the density variations in the quasi-parallel magnetosheath largely come from density variations that are already present in the foreshock and that are amplified when crossing the bow shock (see also our response to the point 2 raised by Reviewer #1). Also, these alternating patches of higher and lower densities in the magnetosheath appear to be associated with irregularities of the shock front, whose scale is comparable to that of the foreshock waves. Previous studies have established that foreshock waves modulate the shape of the shock front [e.g., Burgess, 1995]. Finally, the lower density and velocity variability at lower Mach numbers may be related to the lower amplitude of the foreshock disturbances, or to their smaller scales.

We fully agree with the reviewer that the quasi-parallel shock physics likely also plays an important role in the quasi-parallel magnetosheath, and that bow shock and foreshock effects are hard to disentangle in this global context. In the revised manuscript, we have reworded the ending of the abstract (lines 18-19) and the relevant parts in the discussion and conclusions (lines 426-427, 496, 519, 524-525 and 533) to include quasi-parallel shock physics together with foreshock processes. We have also added more discussion as to how the foreshock affects the quasi-parallel magnetosheath, as detailed just above (lines 426-432).

- Kinetic effects: on 1300 simulation results on density asymmetry are opposed to those coming from an analysis of MHD equations. The authors point to kinetic effects. Why is it that kinetic effects matter specifically on this issue and not on other where simulations and MHD match? This requires more discussion. Even though this may be outside the scope of the paper, a comparison

with 3D MHD simulation (for instance available at CCMC) would help pointing to specific kinetic effects inherent to the Vlasiator code.

We fully agree with the reviewer that it is rather surprising that one of our results regarding the plasma density asymmetry contradict MHD predictions, while a good agreement is found for all other parameters. We thought that this may stem from the fact that foreshock and quasi-parallel shock processes control to a great extent the spatial variations of the density in the quasi-parallel magnetosheath. Because the density asymmetry was more sensitive than the magnetic field strength or the plasma velocity to kinetic processes in the quasi-parallel flank, we argued that kinetic effects might dominate over fluid processes to explain the inconsistency.

However, after reconsidering our quantification of the "global" value of the asymmetries in the different runs, prompted by the first comment of Reviewer #2, we now find that the variation of the density asymmetry with the Mach number is actually inconclusive. Our statement regarding the decrease of the density asymmetry level at low Mach number, which contradicts MHD predictions, was based on the median values of the asymmetry level in the different runs, which were shown to change from -5% (Runs 1 and 2A) to -2% (Run 2B). However, the standard deviations associated with these median values are 5%, 4% and 2% for Runs 1, 2A and 2B, respectively. Also, when comparing visually the curves displayed in Figure 6d-e, there is no evident difference between the different runs, again due to the large variation from bin to bin.

We have therefore reformulated the paragraph at lines 358-366 to state that there is no conclusive difference in the density asymmetry level between the different runs. We thank the reviewer for drawing our attention to this point that helped us resolve the apparent contradiction between MHD and kinetic modelling results.

- Global approach: the model is 2D in space and the magnetopause is not completely resolved such that a model magnetopause needs to be used. This puts limitation on the term "global" to qualify the simulations. I wonder if the compression/expansion in this limited 2D space can be adequately compared with the real 3D situation. Could the authors discuss this aspect? or point to literature as this has surely been already addressed.

We apologize for the lack of clarity regarding the magnetopause description in our simulations. The magnetopause is self-consistently described in our simulation, and its position is determined by pressure balance, just like Earth's magnetopause. A reliable method to evaluate the magnetopause position in numerical simulations is for example based on the magnetosheath flow deflection around the magnetosphere [Palmroth et al., 2003]. However, depending on which criterion is used to define the magnetopause (and the bow shock), the exact position of the boundary thus defined can vary significantly, as the different criteria are not met at exactly the same position [Palmroth et al., 2018, Battarbee et al., 2020]. We have added some clarification regarding the determination of the boundary positions at lines 184-187.

As discussed at lines 364-370 of the initially submitted manuscript, the main consequence of the 2D set-up is the enhanced piling-up of the field lines in front of the magnetopause. This results in a slow expansion of the bow shock and compression of the magnetopause. Therefore, the magnetosheath thickness is somewhat overestimated in the later times of our runs. However, this should not affect the global magnetosheath parameters, except near the magnetopause where the pile-up takes place. In the revised manuscript, we have elaborated on the 2D effects in the discussion (lines 434-436).

- Scales: could the authors give information on the temporal and spatial scales resolved in the simulations? And compare them to typical scales like inertial lengths and typical periods (inverses

of plasma/cyclotron frequencies). How does this compare with the 150s used for averaging magnetosheath parameters? This would help the interpretation of density variability mentioned l289 for instance.

The spatial resolution is 300 km in Run 1 and 228 km in Runs 2A and 2B. The ion inertial length in the solar wind is 228 km in all three runs, which means that we have 1 cell/ion inertial length in Runs 2A and 2B, and 1.3 cell/ion inertial length in Run 1. This resolution is sufficient to resolve ion kinetic processes in a hybrid-Vlasov simulation (see Pfau-Kempf et al., 2018, and our response to Reviewer #1).

The ion cyclotron period in the solar wind is 13 s (Runs 1 and 2A) or 6.5 s (Run 2B). In the magnetosheath, their values are even smaller because of the larger magnetic field strength. The ion plasma period is about 50 ms in the solar wind in all three runs. The 150 s averaging interval used in our study is thus significantly larger than both typical periods, and the variability of the density cannot be linked with the ion gyroperiod for example.

In the revised manuscript, we have added the values of these typical temporal and spatial scales and compared them with the averaging interval (lines 157-162, 175-177, 204-206).

- Set-up: it is not clear to me why run 1 is set up in the XZ plane and arguments are sought for to justify it mimics correctly the XY plane. Why not using a proper set up in the XY plane from the start?

We agree with the reviewer that having all three simulations in the equatorial plane would have been ideal for our study. However, global hybrid-Vlasov simulations are computationally expensive. The runs presented here required from a few million to over 10 million CPU-hours to be carried out. For this study, we decided to make use of the already existing catalogue of Vlasiator simulations that was available to us, and which included runs with upstream conditions that were appropriate for the comparative study we are presenting. Since the different simulation planes are not critical with respect to the magnetosheath properties (provided that the cusp regions are carefully excluded, as we did in Run 1), running a new simulation was not deemed necessary for the present study.

We have added a mention to the computational cost of the simulations in Section 2.1 (lines 171-173), to make it clearer why we use a run in the XZ plane.

- Observations: for comparing observations and simulations the same statistical methodology should be employed, i.e. median or average for both, contrary to what is done in the paper.

Thank you for pointing out this lack of consistency, we should indeed have used the same statistical measure to quantify the "global" value of the asymmetry in each run. In the revised manuscript, we will give the range of values for each asymmetry, rather than the median or the mean which are problematic due to the large variations from bin to bin (see our response to the first point of Reviewer #2 for more detail).

Minor points: - 195: 'warranted'. Do the authors mean 'mandatory'? We will change the wording to "better suited" (line 109).

- Figure 1: mismatch between central / outer legends and d and e labels. We will correct this, thank you for noticing the mismatch.

- 1400: snaller We will correct the typo. - l427: 'statistical'. Do the authors refer to observations here?

Yes, this refers to the observations. We have reformulated this sentence to clarify this (line 516).

Additional references (not previously included in the manuscript bibliography)

Burgess, D.: Foreshock-shock interaction at collisionless quasi-parallel shocks, Advances in Space Research, 15, 159, doi:10.1016/0273-1177(94)00098-L, 1995.

Palmroth, M., Pulkkinen, T. I., Janhunen, P., & Wu, C.-C.: Stormtime energy transfer in global MHD simulation, Journal of Geophysical Research (Space Physics), 108, 1048, doi:10.1029/2002JA009446, 2003.

Pfau-Kempf, Y., Battarbee, M., Ganse, U., Hoilijoki, S., Turc, L., von Alfthan, S., Vainio, R., & Palmroth, M.: On the importance of spatial and velocity resolution in the hybrid-Vlasov modeling of collisionless shocks, Frontiers in Physics, 6, 44, doi:10.3389/fphy.2018.00044, 2018.

**Response to Reviewer #4**

**We thank the referee for their positive evaluation of our manuscript and for providing constructive remarks. Please find below our point-by-point response in **bold** font.**

The paper studies asymmetry in the Earth's dayside magnetosheath using global hybrid-Vlasov simulations and compares numerical results with a statistical dataset of THEMIS observations. The paper is clearly written and the results are new and interesting. However, some details about modeling are missed. I partly agree with the comments of three other reviewers and mention several important points from their reports below. I could recommend the paper for publication after major revision.

**Major remarks**

1. Although the Vlasiator model is well known and I believe it has been thoroughly described in the literature, the paper should provide more details on the runs under discussion. In particular, (as also mentioned by one of the reviewers) the paper says nothing about spatial resolution. It would be useful to compare the resolution with the ion inertial length and gyroradius. The authors have already answered this issue in their reply to Reviewer 1 and I suppose it will appear in the paper too. The paper does not describe the simulation domains in each case; it only mentions that their size is different between the runs. I would be also curious to know what happens if the simulated intervals in Runs 2A and 2B would be increased since now they are shorter than in Run 1.

In Run 1, the simulation box extends from -48.6 to 64  $R_E$  in the x direction and from -59.6 to 39.2  $R_E$  in the z direction. In Runs 2A and 2B, it extends from -7.9 to 46.8  $R_E$  in the x direction and between +/- 31.3  $R_E$  in the y direction. We have added the simulation domain extents, as well as more information regarding the spatial resolution, at lines 157-162 and 175-178.

Our Vlasiator runs comprise two phases. First, in the initialisation phase, the near-Earth magnetic environment forms self-consistently due to the interaction of the dipole field with the incoming solar wind. Then, the run continues in an almost steady state. Due to the 2D set-up of our runs, we never reach a completely steady state because the IMF piles up in front of the magnetopause, causing a slow expansion of the bow shock. In simulations including the foreshock on the dayside, as is the case for the three runs presented here, the main parameter which determines when a run is stopped is when foreshock waves reach the +x boundary, as extending the simulation would likely cause unphysical wave reflection. As concerns the magnetosheath properties, we do not expect significant changes if Runs 2A and 2B were to be extended, except for a larger magnetosheath thickness, due to the field line pile-up.

We now mention that the simulations have reached a quasi-steady state in the interval under study (lines 220-221).

2. Both the Reviewers 1 and 3 noted that comparison with MHD runs for exactly the same solar wind conditions will be useful because this would emphasize which variations in the magnetosheath downstream of the quasi-parallel bow shock are essentially kinetic structures and cannot be predicted by MHD models. However, I do not think that it is necessary to run all MHD models available from CCMC, but it would be enough to make three runs with at least one model (e.g. SWMF/BATSRUS).

While we agree with the reviewer that an in-depth comparison of magnetosheath asymmetries in MHD and kinetic simulations would be an interesting topic of research, we feel that such a study lies beyond the scope of the present paper, as was also noted by Reviewer #3. Identifying the source of discrepancies between different numerical models is not trivial, as many factors can come into play, such as the spatial and temporal resolution, the numerical solvers being used, and so on. Even among MHD models, significant differences are observed, as shown for example by Gordeev et al. [2015], who compared the outputs of the different MHD models available at CCMC.

Our paper presents a comprehensive and self-contained analysis of a set of three hybrid-Vlasov simulations complemented with spacecraft observations, which allow us to draw firm conclusions regarding the effects of several solar wind parameters. Whenever possible, we compared our results with MHD theory and with the MHD simulation results presented in Walsh et al. [2012] and Dimmock et al. [2013], and found them to be in good qualitative agreement. The only apparent discrepancy with MHD theory was the variation of the asymmetry level as a function of the Alfvén Mach number. However, when revisiting those results once the standard deviation of the asymmetry levels was taken into account, based on the suggestion from Reviewer #2, we found that the variation was not conclusive, and thus did not contradict MHD theory (see our response to the third major point of Reviewer #3). We have amended this paragraph (formerly at lines 295-300, now 358-366) when revising the manuscript. For these reasons, we feel that the present paper does not call for an extensive comparison with MHD simulations.

3. I also note that the solar wind conditions in the hybrid simulations are different from the typical solar wind conditions at L1. I am satisfied with the author's reply to Reviewer 1 that the Mach numbers in the solar wind stay in the typical interval and therefore the bow shock-magnetosheath properties may not be changed in comparison with those in observations. However, I would emphasize that the solar wind density of 1 cm-3 is significantly smaller than the average in observations (usually between 5 and 10 cm-3). I think the paper should clearly explain this because I guess that the low solar wind density may be a reason for the stronger fluctuations in the magnetosheath than those in the data.

In the same manner as the high solar wind speed would not influence our results because of the typical Mach numbers in our simulations, the low density should not affect the bow shock-magnetosheath properties either, because the density compression ratio stays within its typical range at Earth. The low solar wind density does not result in large uncertainties in the density in our simulation because the hybrid-Vlasov formalism allows to describe accurately low density plasma, even in regions as tenuous as the magnetotail lobes. This low density does not affect either the development of wave activity in the magnetosheath, which is home to mirror modes [Hoilijoki et al., 2016] and EMIC waves [Dubart et al., 2020].

It would be helpful if the reviewer could provide us with a reference regarding the influence of solar wind density on the variability of magnetosheath properties, as it is not clear to us which other physical processes this could affect. This would be an interesting item to add to the discussion.

4. Since the authors use average parameters both in the simulations and observations, I think it would be possible to add standard deviations to the figures, e.g. in the form of error bars. This would be helpful when comparing the differences between the runs (how significant is the difference with respect to the standard deviations). Besides, the authors mention in the text that they calculated longer time average intervals (line 290). How long are they and does this make any difference to their conclusions?

Thank you for this suggestion. In the revised manuscript, we have added error bars to the asymmetry plots (line plots in Figures 1, 3, 6 and 7). As done in Dimmock et al. [2017], we estimate the error on the magnetosheath parameters as the standard error of the mean (standard deviation divided by the square root of the size of the bin sample). We then use this error to calculate the minimum and maximum values of the asymmetry in each bin, which determines the extent of the error bars in the asymmetry plots. The estimation of the error is described at lines 210-212 and 226-228.

We performed time averages over 50 s, 100 s and 150 s. We did not find any significant differences in the results we obtained. While the exact value of the asymmetry level varied in each bin (especially for the density), the polarity of the asymmetry remained identical, and the range of the asymmetry level over the whole magnetosheath was essentially unchanged. We have added explicitly the duration of the interval on which the averaging was performed to remove the ambiguity in this sentence (lines 352-353).

**Minor remarks**

1. The bibliography list in the paper is long, but I would like to mention two more papers, Zwan and Wolf (https://doi.org/10.1029/JA081i010p01636) who first mentioned the plasma depletion layer and Samsonov et al. (https://doi.org/10.1029/2000JA900150) who compared magnetosheath profiles downstream of the parallel and perpendicular bow shock using the anisotropic MHD model.

**Thank you for these references, we have add them in the introduction.**

2. Line 83. "These processes would thus favour the quasi-parallel flank." But the results in the paper show the Qperp-favoured velocity asymmetry. How is this consistent?

Our results focus on the bulk velocity, which is larger in the quasi-perpendicular magnetosheath. In contrast, the studies of Dimmock et al. [2016a] and Nykyri et al. [2017] show that velocity fluctuations in the Pc 3 range (22 – 100 mHz) are stronger on the quasi-parallel flank and are favourable to the development of the Kelvin-Helmholtz instability. The difference in bulk velocity between the quasi-parallel and quasi-perpendicular flanks is probably not large enough to counteract the effect of the larger velocity fluctuations in the quasi-parallel sector, as it has been shown that the Kelvin-Helmholtz instability is more frequently observed on the quasi-parallel flank [Henry et al., 2017]. We have extend this paragraph of the introduction (lines 93-98) and added the reference to the study by Henry et al. [2017].

3. Lines 149-152. The figures in the paper show that the spatial bins are asymmetric with respect to the Sun-Earth line. Please, explain how this asymmetry is taken into account if you use the same shape as Shue et al.'s model which is symmetrical.

We used a different flaring parameter for the outer magnetosheath boundary on the quasiparallel and quasi-perpendicular flank, to account for the different magnetosheath thicknesses. We have added an explanation for this in the revised manuscript (lines 191-192).

4. Caption to Figure 2. Please, define  $\theta_{Bn}$ .

**We have added the definition of $\theta_{Bn}$ in the figure caption.**

5. Lines 233-235. Is  $\theta_{Bn}$  equal to 0° and 90° near the terminator plane?

**Yes. We have reformulate this sentence to better clarify this (lines 293-294).**

6. Lines 265-266. I think it is better "density compression ratio" instead of "shock compression ratio".

**We agree that "density compression ratio" is less ambiguous. We have corrected this in the revised manuscript.**

7. Label on Figure 5 says that the lines correspond to runs 1 & 2A and 2B but this contradicts the text (lines 265-266).

The text mentions only Runs 2A and 2B, as they are the two runs under discussion at this point in the text. We have added a mention to Run 1 as well to make it clear that the caption and the text are consistent with each other (lines 323-324).

8. Figure 4. The author may add an arrow to indicate the stagnation point.

Thank you for this suggestion. We have added a dashed line in Figure 4 to highlight where the Sun-Earth line is, which shows well that the minima of the velocity curves are shifted towards the quasi-parallel flank.

9. Lines 367-369. Is it better to say about an increase in the magnetic field on the quasi-perpendicular flank than about a decrease on the quasi-parallel flank?

This sentence refers to the low magnetic field strength downstream of the quasi-parallel shock, which remains equally low in the central magnetosheath as in the outer magnetosheath when the cone angle is reduced to  $30^{\circ}$ . In contrast, the magnetic field strength is higher in the central magnetosheath than in the outer magnetosheath for a  $45^{\circ}$  cone angle in Run 1 (see Figure 1a and 1b). On the quasi-perpendicular flank, the magnetic field strength is also lower in Run 2A than in Run 1 because of the lower  $\theta_{Bn}$  value due to the more radial IMF orientation. The field line draping does not cause an increase of the magnetic field strength on the quasi-perpendicular flank in this run. We have reformulated this sentence to clarify this (lines 439-442).

Additional references (not previously included in the manuscript bibliography)

Dubart, M., Ganse, U., Osmane, A., Johlander, A., Battarbee, M., Grandin, M., Pfau-Kempf, Y., Turc, L., and Palmroth, M.: Resolution dependence of magnetosheath waves in global hybrid-Vlasov simulations, Ann. Geophys. Discuss., https://doi.org/10.5194/angeo-2020-24, in review, 2020.

Gordeev, E., Sergeev, V., Honkonen, I., Kuznetsova, M., Rastätter, L., Palmroth, M., Janhunen, P., Tóth, G., Lyon, J., & Wiltberger, M.: Assessing the performance of community-available global MHD models using key system parameters and empirical relationships, Space Weather, 13, 868, doi:10.1002/2015SW001307, 2015.

Hoilijoki, S., Palmroth, M., Walsh, B. M., Pfau-Kempf, Y., von Alfthan, S., Ganse, U., Hannuksela, O., & Vainio, R.: Mirror modes in the Earth's magnetosheath: Results from a global hybrid-Vlasov simulation, Journal of Geophysical Research (Space Physics), 121, 4191, doi:10.1002/2015JA022026, 2016.

Henry, Z. W., Nykyri, K., Moore, T. W., Dimmock, A. P., & Ma, X.: On the Dawn-Dusk Asymmetry of the Kelvin-Helmholtz Instability Between 2007 and 2013, Journal of Geophysical Research (Space Physics), 122, 11,888, doi:10.1002/2017JA024548, 2017.

**Changes made in the manuscript**

**Abstract:**

- A few sentences have been added to better highlight the novelty of our results.
- Quasi-parallel shock effects are now mentioned together with foreshock effects.

**Introduction:**

- We have added references suggested by the reviewers.
- Magnetopause processes are now mentioned as a source of asymmetries for high energy particles.
- We have added a paragraph presenting previous works using kinetic simulations.

**Methodology:**

- We now mention how the plasma moments are calculated from the distribution functions in the simulations.
- Additional information regarding the run parameters is provided (spatial resolution, extent of the spatial domain, Mach numbers, typical temporal scales, etc).
- We provide more details regarding the boundary determination for the magnetosheath binning.
- We describe how the error on the magnetosheath parameters and the asymmetries is estimated.
- We have added a new paragraph at the end of the section to clarify our approach and its validity.

**Results:**

- We removed the mention of the average or the median value of the asymmetry across all azimuthal bins, as it wasn't well representative of the global asymmetry level, and replaced it with the asymmetry level range. This changes in particular the results concerning the density asymmetry, as we now conclude that it does not vary notably from one run to the other.
- We clarified the values of the  $\theta_{Bn}$  angle near the terminator in Run 1.
- We compare the magnetosheath densities we obtained with previous numerical results.

**Discussion:**

- We discuss the effects of the quasi-parallel shock in addition to the foreshock effects.
- We have extended the discussion on possible 2D effects in the simulation.
- We have included a paragraph discussing the spatial resolution in the simulation.
- We have reformulated some sentences to better convey our meaning.

**Conclusions:**

- We have added a few sentences to better highlight the novelty of our results.
- We have reformulated some sentences to better convey our meaning.

**Figures:**

- We have added error bars on Figures 1, 3, 6 and 7
- We have corrected the outer magnetosheath asymmetry in Run 1 (Figures 1e, 3e and 6e) after fixing an error in our analysis program.
- We have added labels to indicated the quasi-parallel and quasi-perpendicular sides and a vertical dashed line in Figure 4.

In addition to the changes requested by the reviewers, we would like to note that we have corrected the results of Run 1 in the outer magnetosheath, as we found an error in the outer boundary parameters, which explained the very irregular density and velocity profiles for this run. This did not change any significant conclusions.

- We modified the discussion on the velocity asymmetry (Section 3.2) based on the updated results
- As a result, we removed from the conclusions the statement that the velocity asymmetry fluctuations are reduced when the Mach number is lower, as this is not shown anymore on the updated figure.
- We modified the description of the density asymmetry in Run 1 accordingly. This did not change any conclusion of the study.

**Asymmetries in the Earth's dayside magnetosheath: results from global hybrid-Vlasov simulations**

Lucile Turc1, Vertti Tarvus1, Andrew Dimmock2, Markus Battarbee1, Urs Ganse1, Andreas Johlander1, Maxime Grandin1, Yann Pfau-Kempf1, Maxime Dubart1, and Minna Palmroth1,3

1Department of Physics, University of Helsinki, Helsinki, Finland
2IRF-U, Uppsala, Sweden

[revised manuscript text omitted]
_{\rm A}$ | $n_{\rm SW} \left[ {\rm cm}^{-3} \right]$ | $\mathbf{V}_{\rm SW}\left[kms^{-1}\right]$ |
|----------|------------------|----------------------|---------------------------------------|-------------------|-------------|-------------------------------------------|--------------------------------------------|
| Run 1    | x-z plane        | 300                  | $45^{\circ}$                          | 5                 | 6.9         | 1                                         | (-750, 0, 0)                               |
| Run 2A   | x - y plane      | 227                  | $30^{\circ}$                          | 5                 | 6.9         | 1                                         | (-750, 0, 0)                               |
| Run 2B   | x - y plane      | 227                  | $30^{\circ}$                          | 10                | 3.5         | 1                                         | (-750, 0, 0)                               |

**2 Methodology**

**2.1 The Vlasiator simulation**

[revised manuscript text omitted]

---

## Referee Report (RR1)

**Review on *Asymmetries in the Earth's dayside magnetosheath: results from global hybrid-Vlasov simulations* by Turc et al.**

The paper has been carefully revised and significantly improved. As I noted in my first review, it contains new results and deserves to be published in Annales Geophysicae. I have only several minor comments.

1. L80-81. The sentence "However, opposite behaviours…" is not directly related to the sentences above. It's better to rephrase.
2. L191-192. Please, provide the flaring parameters for the magnetopause and bow shock.
3. Figure 1d,e. I realize that the error bar is small, but it looks like it is along x rather than y axis. Possibly this is a wrong impression. The authors could mention the obtained errors of the asymmetry in text. Although the way how the authors calculate the errors seems to be reasonable these small errors may look a little strange on zigzag curves like the one showing the density asymmetry (Fig. 6 d,e).
4. L293. The authors state that $\theta_{Bn} \sim 0°$ near terminator in Run 1. But looking at Figure 1, I'm not sure this is correct. I would say $\theta_{Bn} \sim 0°$ for angles of about 50° from the Sun-Earth line.
5. L514-151. The authors explain a stronger asymmetry of the magnetic field in Runs 2a than in Run 1 by the "low $\theta_{Bn}$ near the bow shock nose resulting in a reduced magnetic field compression across most of the quasi-parallel flank of the magnetosheath". (I think it should be a similar statement in Section 3 too.) I wonder if this explanation is consistent with that the asymmetry in Run 2a is also higher near the terminator or even for angles of 45° from the Sun-Earth line. If the authors are convinced that their explanation is correct they could draw a figure similar to Figure 5 but for the ratio of downstream/upstream magnetic field.

---

## Author Response (AR2)

**We thank the reviewer and the editor for their positive evaluation of our manuscript.**

**Our response to the reviewer's comments are in bold in the text below, and the associated edits in the manuscript are also marked in bold fonts.**

The paper has been carefully revised and significantly improved. As I noted in my first review, it contains new results and deserves to be published in Annales Geophysicae. I have only several minor comments.

1.L80-81. The sentence "However, opposite behaviours..." is not directly related to the sentences above. It's better to rephrase.

**We have rephrased this as follows:**

**"It is noteworthy that Paularena et al. [2001] and Dimmock et al. [2016] reported opposite behaviours of the density asymmetry as a function of the solar cycle."**

2.L191-192. Please, provide the flaring parameters for the magnetopause and bow shock.

**We feel that adding a new table with all the different flaring parameters (magnetopause, dusk-side bow shock and dawn-side bow shock for each run) will unnecessarily encumber the paper.**

3.Figure 1d,e. I realize that the error bar is small, but it looks like it is along x rather than y axis. Possibly this is a wrong impression. The authors could mention the obtained errors of the asymmetry in text. Although the way how the authors calculate the errors seems to be reasonable these small errors may look a little strange on zigzag curves like the one showing the density asymmetry (Fig. 6 d,e).

**The error bars are indeed small, and extend along the y axis, and not the x axis. This impression is due to the horizontal bars bounding the error bars (more clearly seen in Figure 7, where the error is significantly larger). The errors are calculated inside each azimuthal bin and the upper and lower bounds are different. We cannot therefore list all error values in the manuscript. Instead, we now give a representative value of the error for the magnetic field asymmetry, which is of the order of 0.1-0.2 %, when the error bars are discussed in the manuscript (at line 260).**

**The error on the density asymmetry is about twice larger (~ 0.4 %) than for the magnetic field asymmetry, but the error bars remain very small.**

4.L293. The authors state that θBn~0° near terminator in Run 1. But looking at Figure 1, I'm not sure this is correct. I would say θBn~0° for angles of about 50° from the Sun-Earth line.

**The figure below shows the θBn angle calculated along the bow shock based on the IMF direction and the normal to a fourth order polynomial fit of the bow shock. As can be seen from this figure, θBn ~ 0 near the terminator in Run 1, as stated in the manuscript. We are now mentioning in the manuscript how θBn was calculated along the bow shock surface, and not determined from visual inspection alone (lines 290-291).**

[Figure]

5.L514-151. The authors explain a stronger asymmetry of the magnetic field in Runs 2a than in Run 1 by the "low θBn near the bow shock nose resulting in a reduced magnetic field compression across most of the quasi-parallel flank of the magnetosheath". (I think it should be a similar statement in Section 3 too.) I wonder if this explanation is consistent with that the asymmetry in Run 2a is also higher near the terminator or even for angles of 45° from the Sun-Earth line. If the authors are convinced that their explanation is correct they could draw a figure similar to Figure 5 but for the ratio of downstream/upstream magnetic field.

**Indeed, purely based on Rankine-Hugoniot relations, the asymmetry just downstream of the bow shock should be stronger near the terminator in Run 1 than in Run 2A. However, in our work, the asymmetry is not calculated just downstream of the bow shock, but in a sizeable fraction of the magnetosheath. Therefore, one must also take into account the magnetosheath flow and where the streamlines are connected to the bow shock. A large part of the plasma populating the magnetosheath near the terminator actually originates from parts of the bow shock located much closer to the subsolar point. This likely explains why the asymmetry remains larger in Run 1 than in Run 2A away from the subsolar point.**

**We now mention this effect at lines 272-274.**

[revised manuscript text omitted]